# ENSEMBLE DISTRIBUTION DISTILLATION

**Andrey Malinin** [*]
Yandex
am969@yandex-team.ru

**Bruno Mlodozeniec** [*]
Department of Engineering
University of Cambridge
bkm28@cam.ac.uk

**Mark Gales**
Department of Engineering
University of Cambridge
mjfg@eng.cam.ac.uk

## ABSTRACT

Ensembles of models often yield improvements in system performance. These ensemble approaches have also been empirically shown to yield robust measures of uncertainty, and are capable of distinguishing between different *forms* of uncertainty. However, ensembles come at a computational and memory cost which may be prohibitive for many applications. There has been significant work done on the distillation of an ensemble into a single model. Such approaches decrease computational cost and allow a single model to achieve an accuracy comparable to that of an ensemble. However, information about the *diversity* of the ensemble, which can yield estimates of different forms of uncertainty, is lost. This work considers the novel task of *Ensemble Distribution Distillation* (EnD$^2$) — distilling the distribution of the predictions from an ensemble, rather than just the average prediction, into a single model. EnD$^2$ enables a single model to retain both the improved classification performance of ensemble distillation as well as information about the diversity of the ensemble, which is useful for uncertainty estimation. A solution for EnD$^2$ based on Prior Networks, a class of models which allow a single neural network to explicitly model a distribution over output distributions, is proposed in this work. The properties of EnD$^2$ are investigated on both an artificial dataset, and on the CIFAR-10, CIFAR-100 and TinyImageNet datasets, where it is shown that EnD$^2$ can approach the classification performance of an ensemble, and outperforms both standard DNNs and Ensemble Distillation on the tasks of misclassification and out-of-distribution input detection.

## 1 INTRODUCTION

Neural Networks (NNs) have emerged as the state-of-the-art approach to a variety of machine learning tasks (LeCun et al., 2015) in domains such as computer vision (Girshick, 2015; Simonyan & Zisserman, 2015; Villegas et al., 2017), natural language processing (Mikolov et al., 2013b;a; 2010), speech recognition (Hinton et al., 2012; Hannun et al., 2014) and bio-informatics (Caruana et al., 2015; Alipanahi et al., 2015). Despite impressive supervised learning performance, NNs tend to make over-confident predictions (Lakshminarayanan et al., 2017) and, until recently, have been unable to provide measures of uncertainty in their predictions. As NNs are increasingly being applied to safety-critical tasks such as medical diagnosis (De Fauw et al., 2018), biometric identification (Schroff et al., 2015) and self driving cars, estimating uncertainty in model's predictions is crucial, as it enables the safety of an AI system (Amodei et al., 2016) to be improved by acting on the predictions in an informed manner.

Ensembles of NNs are known to yield increased accuracy over a single model (Murphy, 2012), allow useful measures of uncertainty to be derived (Lakshminarayanan et al., 2017), and also provide defense against adversarial attacks (Smith & Gal, 2018). There is both a range of Bayesian Monte-Carlo approaches (Gal & Ghahramani, 2016; Welling & Teh, 2011; Garipov et al., 2018; Maddox et al., 2019), as well as non-Bayesian approaches, such as random-initialization (Lakshminarayanan et al., 2017) and bagging (Murphy, 2012; Osband et al., 2016), to generating ensembles. Crucially, ensemble approaches allow total uncertainty in predictions to be decomposed into *knowledge uncertainty* and *data uncertainty*. *Data uncertainty* is the irreducible uncertainty in predictions which arises due to the complexity, multi-modality and noise in the data. *Knowledge uncertainty*, also known as *epistemic*

---

[*]Equal Contribution

*uncertainty* (Gal, 2016) or *distributional uncertainty* (Malinin & Gales, 2018), is uncertainty due to a *lack of understanding* or *knowledge* on the part of the model regarding the current input for which the model is making a prediction. This form of uncertainty arises when the *test input $\boldsymbol{x}^*$* comes either from a different distribution than the one that generated the training data or from an in-domain region which is sparsely covered by the training data. Mismatch between the test and training distributions is also known as a dataset shift (Quiñonero-Candela, 2009), and is a situation which often arises for real world problems. Distinguishing between sources of uncertainty is important, as in certain machine learning applications it may be necessary to know not only *whether* the model is uncertain, but also *why*. For instance, in active learning, additional training data should be collected from regions with high *knowledge uncertainty*, but not *data uncertainty*.

A fundamental limitation of ensembles is that the computational cost of training and, more importantly, inference can be many times greater than that of a single model. One solution to speed up inference is to distill an ensemble of models into a single network to yield the mean predictions of the ensemble (Hinton et al., 2015; Korattikara Balan et al., 2015). However, this collapses an ensemble of conditional distributions over classes into a single point-estimate conditional distribution over classes. As a result, information about the *diversity* of the ensemble is lost. This prevents measures of *knowledge uncertainty*, such as mutual information (Malinin & Gales, 2018; Depeweg et al., 2017a), from being estimated.

In this work, we investigate the explicit modelling of the distribution over the ensemble predictions, rather than just the mean, with a single model. This problem — referred to as *Ensemble Distribution Distillation* (EnD$^2$) — yields a method that preserves *both* the distributional information and improved classification performance of an ensemble within a *single* neural network model. It is important to highlight that *Ensemble Distribution Distillation* is a **novel task** which, to our knowledge, has not been previously investigated. Here, the goal is to extract as much information as possible from an ensemble of models and retain it within a single, possibly simpler, model. As an initial solution to this problem, this paper makes use of a recently introduced class of models, known as *Prior Networks* (Malinin & Gales, 2018; 2019), which explicitly model a conditional *distribution over categorical distributions* by parameterizing a *Dirichlet distribution*. Within the context of EnD$^2$ this effectively allows a single model to *emulate* the complete ensemble.

The contributions of this work are as follows. Firstly, we define the task of Ensemble Distribution Distillation (EnD$^2$) as a new challenge for machine learning research. Secondly, we propose and evaluate a solution to this problem using Prior Networks. EnD$^2$ is initially investigated on artificial data, which allows the behaviour of the models to be visualized. It is shown that distribution-distilled models are able to distinguish between *data uncertainty* and *knowledge uncertainty*. Finally, EnD$^2$ is evaluated on CIFAR-10, CIFAR-100 and TinyImageNet datasets, where it is shown that EnD$^2$ yields models which approach the classification performance of the original ensemble and outperform standard DNNs and regular Ensemble Distillation (EnD) models on the tasks of identifying misclassifications and out-of-distribution (OOD) samples.

## 2 ENSEMBLES

In this work, a *Bayesian* viewpoint on ensembles is adopted, as it provides a particularly elegant probabilistic framework, which allows *knowledge uncertainty* to be linked to Bayesian *model uncertainty*. However, it is also possible to construct ensembles using a range of non-Bayesian approaches. For example, it is possible to *explicitly* construct an ensemble of $M$ models by training on the same data with different random seeds (Lakshminarayanan et al., 2017) and/or different model architectures. Alternatively, it is possible to generate ensembles via *Bootstrap* methods (Murphy, 2012; Osband et al., 2016) in which each model is trained on a re-sampled version of the training data.

The essence of Bayesian methods is to treat the model parameters $\boldsymbol{\theta}$ as random variables and place a prior distribution $\mathrm{p}(\boldsymbol{\theta})$ over them to compute a posterior distribution $\mathrm{p}(\boldsymbol{\theta}|\mathcal{D})$ via Bayes' rule:

$$\mathrm{p}(\boldsymbol{\theta}|\mathcal{D}) = \frac{\mathrm{p}(\mathcal{D}|\boldsymbol{\theta})\mathrm{p}(\boldsymbol{\theta})}{\mathrm{p}(\mathcal{D})} \propto \mathrm{p}(\mathcal{D}|\boldsymbol{\theta})\mathrm{p}(\boldsymbol{\theta}) \tag{1}$$

Here, *model uncertainty* is captured in the posterior distribution $\mathrm{p}(\boldsymbol{\theta}|\mathcal{D})$. Consider an ensemble of models $\{\mathrm{P}(y|\boldsymbol{x}^*, \boldsymbol{\theta}^{(m)})\}_{m=1}^M$ sampled from the posterior:

$$\left\{\mathrm{P}(y|\boldsymbol{x}^*, \boldsymbol{\theta}^{(m)})\right\}_{m=1}^M \rightarrow \left\{\mathrm{P}(y|\boldsymbol{\pi}^{(m)})\right\}_{m=1}^M, \quad \boldsymbol{\pi}^{(m)} = \boldsymbol{f}(\boldsymbol{x}^*; \boldsymbol{\theta}^{(m)}), \ \boldsymbol{\theta}^{(m)} \sim \mathrm{p}(\boldsymbol{\theta}|\mathcal{D}) \tag{2}$$

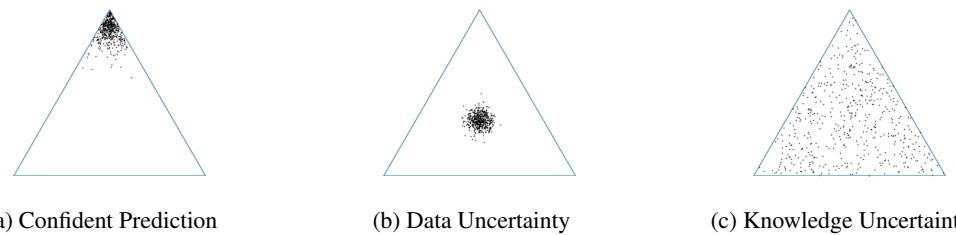

(a) Confident Prediction        (b) Data Uncertainty        (c) Knowledge Uncertainty

Figure 1: Desired behaviors of a ensemble on a simplex of categorical probabilities.

where $\boldsymbol{\pi}$ are the parameters of a categorical distribution $[\mathrm{P}(y = \omega_1), \cdots , \mathrm{P}(y = \omega_K)]^{\mathsf{T}}$ and $\boldsymbol{x}^*$ is a *test input*. The expected predictive distribution, or *predictive posterior*, for a test input $\boldsymbol{x}^*$ is obtained by taking the expectation with respect to the model posterior:

$$\mathrm{P}(y|\boldsymbol{x}^*, \mathcal{D}) = \mathbb{E}_{\mathrm{p}(\boldsymbol{\theta}|\mathcal{D})}\big[\mathrm{P}(y|\boldsymbol{x}^*, \boldsymbol{\theta})\big] \tag{3}$$

Each of the models $\mathrm{P}(y|\boldsymbol{x}^*, \boldsymbol{\theta}^{(m)})$ yields a *different* estimate of *data uncertainty*. Uncertainty in predictions due to *model uncertainty* is expressed as the level of spread, or 'disagreement', of an ensemble sampled from the posterior. The aim is to craft a posterior $\mathrm{p}(\boldsymbol{\theta}|\mathcal{D})$, via an appropriate choice of prior $\mathrm{p}(\boldsymbol{\theta})$, which yields an ensemble that exhibits the set of behaviours described in figure 1. Specifically, for an in-domain test input $\boldsymbol{x}^*$, the ensemble should produce a consistent set of predictions with little spread, as described in figure 1a and figure 1b. In other words, the models should agree in their estimates of *data uncertainty*. On the other hand, for inputs which are different from the training data, the models in the ensemble should 'disagree' and produce a diverse set of predictions, as shown in figure 1c. Ideally, the models should yield increasingly diverse predictions as input $\boldsymbol{x}^*$ moves further away from the training data. If an input is completely unlike the training data, then the level of disagreement should be significant. Hence, the measures of *model uncertainty* will capture *knowledge uncertainty* given an appropriate choice of prior.

Given an ensemble $\big\{\mathrm{P}(y|\boldsymbol{x}^*, \boldsymbol{\theta}^{(m)})\big\}_{m=1}^{M}$ which exhibits the desired set of behaviours, the entropy of the expected distribution $\mathrm{P}(y|\boldsymbol{x}^*, \mathcal{D})$ can be used as a measure of *total uncertainty* in the prediction. Uncertainty in predictions due to *knowledge uncertainty* can be assessed via measures of the spread, or 'disagreement', of the ensemble such as *Mutual Information*:

$$\underbrace{\mathcal{MI}[y, \boldsymbol{\theta}|\boldsymbol{x}^*, \mathcal{D}]}_{Knowledge\ Uncertainty} = \underbrace{\mathcal{H}\big[\mathbb{E}_{\mathrm{p}(\boldsymbol{\theta}|\mathcal{D})}[\mathrm{P}(y|\boldsymbol{x}^*, \boldsymbol{\theta})]\big]}_{Total\ Uncertainty} - \underbrace{\mathbb{E}_{\mathrm{p}(\boldsymbol{\theta}|\mathcal{D})}\big[\mathcal{H}[\mathrm{P}(y|\boldsymbol{x}^*, \boldsymbol{\theta})]\big]}_{Expected\ Data\ Uncertainty} \tag{4}$$

This formulation of mutual information allows the *total uncertainty* to be decomposed into *knowledge uncertainty* and *expected data uncertainty* (Depeweg et al., 2017a;b). The entropy of the predictive posterior, or *total uncertainty*, will be high whenever the model is uncertain - both in regions of severe class overlap and out-of-domain. However, the difference of the entropy of the predictive posterior and the expected entropy of the individual models will be non-zero only if the models disagree. For example, in regions of class overlap, *each* member of the ensemble will yield a high entropy distribution (figure 1b) - the entropy of the predictive posterior and the expected entropy will be similar and mutual information will be low. In this situation *total uncertainty* is dominated by *data uncertainty*. On the other hand, for out-of-domain inputs the ensemble yields diverse distributions over classes such that the predictive posterior is near uniform (figure 1c), while the expected entropy of each model may be much lower. In this region of input space the models' understanding of data is low and, therefore, *knowledge uncertainty* is high.

## 3   ENSEMBLE DISTRIBUTION DISTILLATION

Previous work (Hinton et al., 2015; Korattikara Balan et al., 2015; Wong & Gales, 2017; Wang et al., 2018; Papamakarios, 2015; Buciluǎ et al., 2006) has investigated *distilling* a single large network into a smaller one and an ensemble of networks into a single neural network. In general, distillation is done by minimizing the KL-divergence between the model and the expected predictive distribution of an ensemble:

$$\mathcal{L}(\boldsymbol{\phi}, \mathcal{D}_{\mathrm{ens}}) = \mathbb{E}_{\hat{\mathrm{p}}(\boldsymbol{x})}\Big[\mathrm{KL}\big[\mathbb{E}_{\hat{\mathrm{p}}(\boldsymbol{\theta}|\mathcal{D})}[\mathrm{P}(y|\boldsymbol{x}; \boldsymbol{\theta})] \,||\, \mathrm{P}(y|\boldsymbol{x}; \boldsymbol{\phi})]\big]\Big] \tag{5}$$

This approach essentially aims to train a single model that captures the *mean* of an ensemble, allowing the model to achieve a higher classification performance at a far lower computational cost. The use of such models for uncertainty estimation was investigated in (Li & Hoiem, 2019; Englesson & Azizpour, 2019). However, the limitation of this approach with regards to uncertainty estimation is that the information about the *diversity* of the ensemble is lost. As a result, it is no longer possible to decompose *total uncertainty* into *knowledge uncertainty* and *data uncertainty* via mutual information as in equation 4. In this work we propose the task of *Ensemble Distribution Distillation*, where the goal is to capture not only the mean of the ensemble, but also its diversity. In this section, we outline an initial solution to this task.

An ensemble can be viewed as a set of samples from an *implicit* distribution of output distributions:

$$\big\{\mathrm{P}(y|\boldsymbol{x}^*, \boldsymbol{\theta}^{(m)})\big\}_{m=1}^M \rightarrow \big\{\mathrm{P}(y|\boldsymbol{\pi}^{(m)})\big\}_{m=1}^M, \quad \boldsymbol{\pi}^{(m)} \sim \mathrm{p}(\boldsymbol{\pi}|\boldsymbol{x}^*, \mathcal{D}) \tag{6}$$

Recently, a new class of models was proposed, called *Prior Networks* (Malinin & Gales, 2018; 2019), which *explicitly* parameterize a conditional distribution over output distributions $\mathrm{p}(\boldsymbol{\pi}|\boldsymbol{x}^*; \hat{\boldsymbol{\phi}})$ using a single neural network parameterized by a point estimate of the model parameters $\hat{\boldsymbol{\phi}}$. Thus, a Prior Network is able to effectively *emulate* an ensemble, and therefore yield the same measures of uncertainty. A Prior Network $\mathrm{p}(\boldsymbol{\pi}|\boldsymbol{x}^*; \hat{\boldsymbol{\phi}})$ models a distribution over categorical output distributions by parameterizing the Dirichlet distribution.

$$\mathrm{p}(\boldsymbol{\pi}|\boldsymbol{x}; \hat{\boldsymbol{\phi}}) = \mathrm{Dir}(\boldsymbol{\pi}|\hat{\boldsymbol{\alpha}}), \quad \hat{\boldsymbol{\alpha}} = \boldsymbol{f}(\boldsymbol{x}; \hat{\boldsymbol{\phi}}), \quad \hat{\alpha}_c > 0, \ \hat{\alpha}_0 = \sum_{c=1}^K \hat{\alpha}_c \tag{7}$$

The distribution is parameterized by its concentration parameters $\boldsymbol{\alpha}$, which can be obtained by placing an exponential function at the output of a Prior Network: $\hat{\alpha}_c = e^{\hat{z}_c}$, where $\hat{\boldsymbol{z}}$ are the logits predicted by the model. While a Prior Network could, in general, parameterize arbitrary distributions over categorical distributions, the Dirichlet is chosen due to its tractable analytic properties, which allow closed form expressions for all measures of uncertainty to be obtained. However, it is important to note that the Dirichlet distribution may be too limited to fully capture the behaviour of an ensemble and other distributions may need to be considered.

In this work we consider how an ensemble, which is a set of samples from an *implicit* distribution over distributions, can be *distribution distilled* into an *explicit* distribution over distributions modelled using a single Prior Network model, ie: $\big\{\mathrm{P}(y|\boldsymbol{x}; \boldsymbol{\theta}^{(m)})\big\}_{m=1}^M \rightarrow \mathrm{p}(\boldsymbol{\pi}|\boldsymbol{x}; \hat{\boldsymbol{\phi}})$.

This is accomplished in several steps. Firstly, a *transfer dataset* $\mathcal{D}_{\mathrm{ens}} = \big\{\boldsymbol{x}^{(i)}, \boldsymbol{\pi}^{(i,1:M)}\big\}_{i=1}^N \sim \hat{\mathrm{p}}(\boldsymbol{x}, \boldsymbol{\pi})$ is composed of the inputs $\boldsymbol{x}_i$ from the original training set $\mathcal{D} = \{\boldsymbol{x}^{(i)}, y^{(i)}\}_{i=1}^N$ and the categorical distributions $\{\boldsymbol{\pi}^{(i,1:M)}\}_{i=1}^N$ derived from the ensemble for each input. Secondly, given this transfer set, the model $\mathrm{p}(\boldsymbol{\pi}|\boldsymbol{x}; \phi)$ is trained by minimizing the negative log-likelihood of each categorical distribution $\boldsymbol{\pi}^{(im)}$:

$$\mathcal{L}(\boldsymbol{\phi}, \mathcal{D}_{\mathrm{ens}}) = -\mathbb{E}_{\hat{\mathrm{p}}(\boldsymbol{x})}\big[\mathbb{E}_{\hat{\mathrm{p}}(\boldsymbol{\pi}|\boldsymbol{x})}[\ln \mathrm{p}(\boldsymbol{\pi}|\boldsymbol{x}; \phi)]\big]$$
$$= -\frac{1}{N}\sum_{i=1}^N \bigg[\ln\Gamma(\hat{\alpha}_0^{(i)}) - \sum_{c=1}^K \ln\Gamma(\hat{\alpha}_c^{(i)}) + \frac{1}{M}\sum_{m=1}^M\sum_{c=1}^K (\hat{\alpha}_c^{(i)} - 1)\ln\pi_c^{(im)}\bigg] \tag{8}$$

Thus, Ensemble Distribution Distillation with Prior Networks is a straightforward application of maximum-likelihood estimation. Given a distribution-distilled Prior Network, the predictive distribution is given by the expected categorical distribution $\hat{\boldsymbol{\pi}}$ under the Dirichlet prior:

$$\mathrm{P}(y = \omega_c|\boldsymbol{x}^*; \hat{\boldsymbol{\phi}}) = \mathbb{E}_{\mathrm{p}(\boldsymbol{\pi}|\boldsymbol{x}^*; \hat{\boldsymbol{\phi}})}[\mathrm{P}(y = \omega_c|\boldsymbol{\pi})] = \hat{\pi}_c = \frac{\hat{\alpha}_c}{\sum_{k=1}^K \hat{\alpha}_k} = \frac{e^{\hat{z}_c}}{\sum_{k=1}^K e^{\hat{z}_k}} \tag{9}$$

Separable measures of uncertainty can be obtained by considering the mutual information between the prediction $y$ and the parameters of $\boldsymbol{\pi}$ of the categorical:

$$\underbrace{\mathcal{MI}[y, \boldsymbol{\pi}|\boldsymbol{x}^*; \hat{\boldsymbol{\phi}}]}_{Knowledge\ Uncertainty} = \underbrace{\mathcal{H}\big[\mathbb{E}_{\mathrm{p}(\boldsymbol{\pi}|\boldsymbol{x}^*; \hat{\boldsymbol{\phi}})}[\mathrm{P}(y|\boldsymbol{\pi})]\big]}_{Total\ Uncertainty} - \underbrace{\mathbb{E}_{\mathrm{p}(\boldsymbol{\pi}|\boldsymbol{x}^*; \hat{\boldsymbol{\phi}})}\big[\mathcal{H}[\mathrm{P}(y|\boldsymbol{\pi})]\big]}_{Expected\ Data\ Uncertainty} \tag{10}$$

Similar to equation 4, this expression allows *total uncertainty*, given by the entropy of the expected distribution, to be decomposed into *data uncertainty* and *knowledge uncertainty* (Malinin & Gales, 2018). If Ensemble Distribution Distillation is successful, then the measures of uncertainty derivable from a distribution-distilled model should be identical to those derived from the original ensemble.

## 3.1 TEMPERATURE ANNEALING

Minimization of the negative log-likelihood of the model on the transfer dataset $\mathcal{D}_{\text{ens}} = \{x^{(i)}, \pi^{(i,1:M)}\}_{i=1}^N$ is equivalent to minimization of the KL-divergence between the model and the empirical distribution $\hat{p}(x, \pi)$. On training data, this distribution is often 'sharp' at one of the corners of the simplex. At the same time, the Dirichlet distribution predicted by the model has its mode near the center of the simplex with little support at the corners at initialization. Thus, the common support between the model and the target empirical distribution is limited. Optimization of the KL-divergence between distributions with limited non-zero common support is particularly difficult. To alleviate this issue, and improve convergence, the proposed solution is to use *temperature* to 'heat up' both distributions and increase common support by moving the modes of both distributions closer together. The empirical distribution is 'heated up' by raising the temperature $T$ of the softmax of each model in the ensemble in the same way as in (Hinton et al., 2015). This moves the predictions of the ensemble closer to the center of the simplex and *decreases* their diversity, making it better modelled by a sharp Dirichlet distribution. The output distribution of the EnD$^2$ model $p(\pi|x; \phi)$ is heated up by raising the temperature of the concentration parameters: $\hat{\alpha}_c = e^{\hat{z}_c/T}$, making support more uniform across the simplex. An *annealing schedule* is used to re-emphasize the diversity of the empirical distribution and return it to its 'natural' state by lowering the temperature down to 1 as training progresses.

## 4 EXPERIMENTS ON ARTIFICIAL DATA

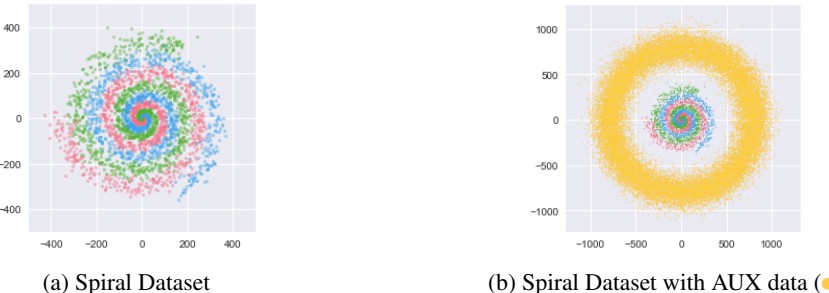

(a) Spiral Dataset  (b) Spiral Dataset with AUX data (●)

Figure 2: 3-spiral dataset with 1000 examples per class

The current section investigates Ensemble Distribution Distillation (EnD$^2$) on an artificial dataset shown in figure 2a. This dataset consists of three spiral arms extending from the center with both increasing noise and distance between the arms. Each arm corresponds to a single class. This dataset is chosen such that it is *not* linearly separable and requires a powerful model to correctly model the decision boundaries, and also such that there are definite regions of class overlap.

In the following set of experiments, an ensemble of 100 neural networks is constructed by training neural networks from 100 different random initializations. A smaller (sub) ensemble of only 10 neural networks is also considered. The models are trained on 3000 data-points sampled from the spiral dataset, with 1000 examples per class. The classification performance of EnD$^2$ is compared to the performance of individual neural networks, the overall ensemble and regular Ensemble Distillation (EnD). The results are presented in table 1.

Table 1: Classification Performance (% Error) on $\mathcal{D}_{test}$ of size 1000, trained on $\mathcal{D}_{trn}$ of size 1000 with 3 spiral classes. Dataset sizes given as number of examples per class.

| Num. models | Individual | Ensemble | EnD | EnD$^2$ |
|:---:|:---:|:---:|:---:|:---:|
| 10 | | 12.63 | 12.57 | 12.52 |
| 100 | 13.21 | 12.37 | 12.39 | 12.47 |

The results show that an ensemble of 10 models has a clear performance gain compared to the mean performance of the individual models. An ensemble of 100 models has a smaller performance gain

over an ensemble of only 10 models. Ensemble Distillation (EnD) is able to recover the classification performance of both an ensemble of 10 and 100 models with only very minor degradation in performance. Finally, Ensemble Distribution Distillation is also able to recover most of the performance gain of an ensemble, but with a slightly larger degradation. This is likely due to forcing a single model to learn not only the mean, but also the distribution around it, which likely requires more capacity from the network. The measures of uncertainty derived form an ensemble of 100 models and

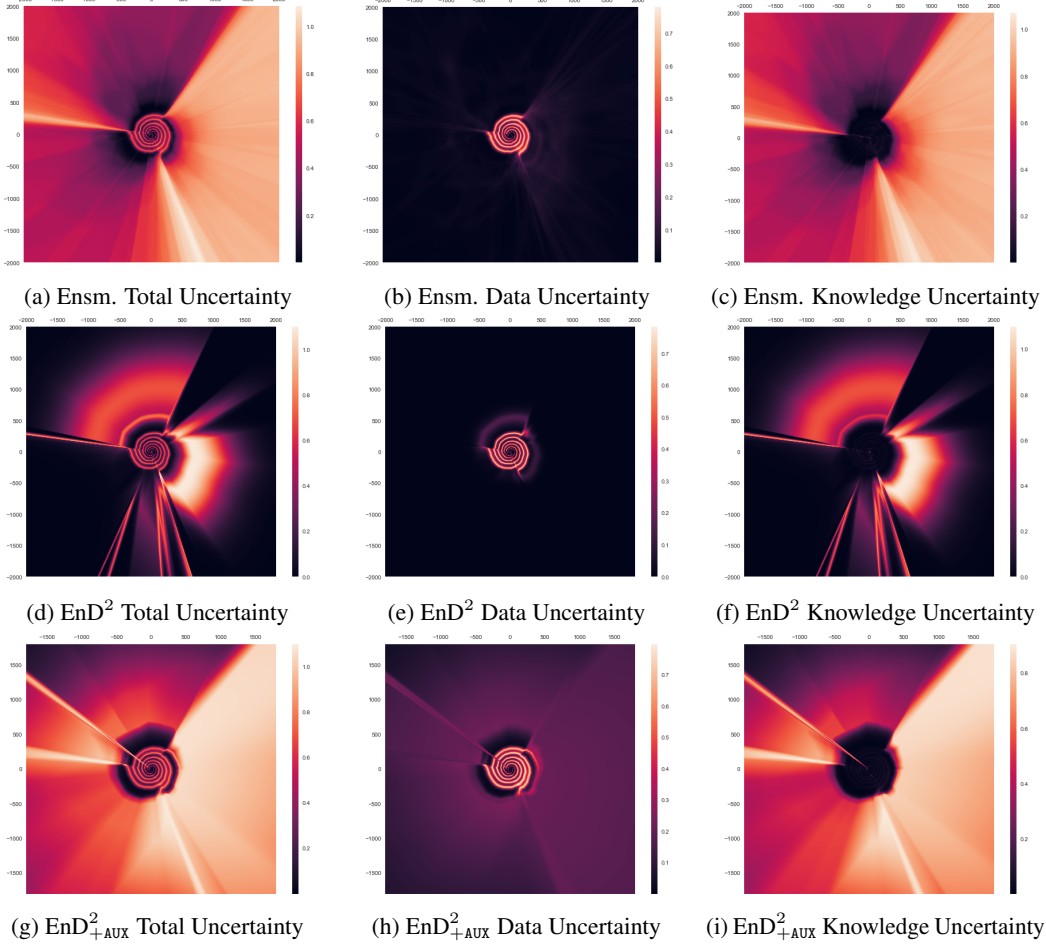

(a) Ensm. Total Uncertainty  (b) Ensm. Data Uncertainty  (c) Ensm. Knowledge Uncertainty

(d) EnD$^2$ Total Uncertainty  (e) EnD$^2$ Data Uncertainty  (f) EnD$^2$ Knowledge Uncertainty

(g) EnD$^2_{+\text{AUX}}$ Total Uncertainty  (h) EnD$^2_{+\text{AUX}}$ Data Uncertainty  (i) EnD$^2_{+\text{AUX}}$ Knowledge Uncertainty

Figure 3: Comparison of measures of uncertainty derived from an Ensemble, EnD$^2$ and EnD$^2_{+\text{AUX}}$.

from Ensemble Distribution Distillation are presented in figures 3a-c and figures 3d-f, respectively. The results show that EnD$^2$ successfully captures data uncertainty and also correctly decomposes *total uncertainty* into *knowledge uncertainty* and *data uncertainty*. However, it fails to appropriately capture *knowledge uncertainty* further away from the training region, as there are obvious dark holes in figure 3f, where the model yields low *knowledge uncertainty* far from the region of training data.

In order to overcome these issues, a thick ring of inputs far from the training data was sampled as depicted in figure 2b. The predictions of the ensemble were obtained for these input points and used as additional *auxiliary* training data $\mathcal{D}_{\text{AUX}}$. Table 2 shows how using the auxiliary training data affects the performance of the Ensemble Distillation and Ensemble Distribution Distillation. There is a minor drop in performance of both distillation approaches. However, the overall level of performance is not compromised and is still higher than the average performance of each individual DNN model. The behaviour of measures of uncertainty derived from Ensemble Distribution Distillation with auxiliary training data (EnD$^2_{+\text{AUX}}$) is shown in figures 3g-i. These results show that successful Ensemble Distribution Distillation of the *out-of-distribution behaviour* of an ensemble based purely on observations of the *in-domain behaviour* is challenging and may require the use of additional

training data. This is compounded by the fact that the diversity of an ensemble on *training* data that the model has seen is typically smaller than on a heldout test-set.

Table 2: Classification Performance (% Error) on $\mathcal{D}_{test}$, trained on either $\mathcal{D}_{trn}$ or $\mathcal{D}_{trn} + \mathcal{D}_{\texttt{AUX}}$. All datasets are of size 1000. Data for an ensemble of a 100 models.

| Distillation Data | Individual | Ensemble | EnD | EnD$^2$ |
|---|---|---|---|---|
| $\mathcal{D}_{\texttt{trn}}$ | 13.21 | 12.37 | 12.39 | 12.47 |
| $\mathcal{D}_{\texttt{trn}} + \mathcal{D}_{\texttt{AUX}}$ | | | 12.41 | 12.50 |

## 5 EXPERIMENTS ON IMAGE DATA

Having confirmed the properties of EnD$^2$ on an artificial dataset, we now investigate Ensemble Distribution Distillation on the CIFAR-10 (C10), CIFAR-100 (C100) and TinyImageNet (TIM) (Krizhevsky, 2009; CS231N, 2017) datasets. Similarly to section 4, an ensemble of a 100 models is constructed by training NNs on C10/100/TIM data from different random initializations. The *transfer dataset* is constructed from C10/100/TIM inputs and ensemble logits to allow for *temperature annealing* during training, which we found to be essential to getting EnD$^2$ to train well. In addition, we also consider Ensemble Distillation and Ensemble Distribution Distillation on a transfer set that contains both the original C10/C100/TIM training data and *auxiliary* (AUX) data taken from the other dataset[1], termed EnD$_{+\texttt{AUX}}$ and EnD$^2_{+\texttt{AUX}}$ respectively. It is important to note that the auxiliary data has been treated in the same way as main data during construction of the transfer set and distillation. This offers an advantage over traditional Prior Network training (Malinin & Gales, 2018; 2019), where the knowledge of which examples are in-domain and out-of-distribution is required a-priori. In this work we also make a comparison with Prior Networks trained via reverse KL-divergence (Malinin & Gales, 2019), where the Prior Networks (PN) are trained on the same datasets, both main and auxiliary, as the EnD$_{+AUX}$ and EnD$^2_{+AUX}$ models[2]. Note, that for Ensemble Distribution Distillation, models can be distribution-distilled using *any* (potentially unlabeled) auxiliary data on which ensemble predictions can be obtained. Further note that in these experiments we explicitly chose to use the simpler VGG-16 (Simonyan & Zisserman, 2015) architecture rather than more modern architectures like ResNet (He et al., 2016) as the goal of this work is to analyse the properties of Ensemble Distribution Distillation in a clean and simple configuration. The datasets and training configurations for all models are detailed in appendix A.

Table 3: Mean Classification Error, % PRR , test-set negative log-likelihood (NLL) and expected calibration error (ECE) on C10/C100/TIM across three models $\pm 2\sigma$.

| DSET | CRIT. | IND | ENSM | EnD | EnD$^2$ | EnD$_{+\texttt{AUX}}$ | EnD$^2_{+\texttt{AUX}}$ | PN$_{+\texttt{AUX}}$ |
|---|---|---|---|---|---|---|---|---|
| C10 | ERR | 8.0 $\pm0.4$ | **6.2** $\pm$ NA | 6.7 $\pm0.3$ | 7.3 $\pm0.2$ | 6.7 $\pm0.2$ | 6.9 $\pm0.2$ | 7.5 $\pm0.3$ |
| | PRR | 84.6 $\pm1.2$ | **86.8** $\pm$ NA | 84.8 $\pm0.8$ | 85.3 $\pm1.1$ | 85.1 $\pm0.1$ | 85.7 $\pm0.3$ | 82.0 $\pm1.4$ |
| | ECE | 2.2 $\pm0.4$ | 1.3 $\pm$ NA | 2.6 $\pm0.2$ | **1.0** $\pm0.2$ | 2.6 $\pm0.6$ | 2.2 $\pm0.4$ | 12.0 $\pm0.7$ |
| | NLL | 0.25 $\pm0.01$ | **0.19** $\pm$ NA | 0.22 $\pm0.01$ | 0.25 $\pm0.01$ | 0.22 $\pm0.01$ | 0.24 $\pm0.00$ | 0.38 $\pm0.01$ |
| C100 | ERR | 30.4 $\pm0.3$ | **26.3** $\pm$ NA | 28.0 $\pm0.4$ | 27.9 $\pm0.3$ | 28.2 $\pm0.3$ | 28.0 $\pm0.5$ | 28.0 $\pm0.7$ |
| | PRR | 72.5 $\pm1.0$ | **75.0** $\pm$ NA | 73.1 $\pm0.5$ | 73.7 $\pm0.7$ | 74.0 $\pm0.3$ | 74.0 $\pm0.2$ | 63.7 $\pm0.8$ |
| | ECE | 9.3 $\pm0.8$ | **1.2** $\pm$ NA | 8.2 $\pm0.3$ | 4.9 $\pm0.5$ | 1.9 $\pm0.3$ | 5.6 $\pm0.5$ | 37.9 $\pm0.4$ |
| | NLL | 1.16 $\pm0.03$ | **0.88** $\pm$ NA | 1.06 $\pm0.01$ | 1.14 $\pm0.01$ | 0.98 $\pm0.00$ | 1.14 $\pm0.01$ | 1.87 $\pm0.03$ |
| TIM | ERR | 41.8 $\pm0.6$ | **36.6** $\pm$NA | 38.3 $\pm0.2$ | 37.6 $\pm0.2$ | 38.5 $\pm0.3$ | 37.3 $\pm0.5$ | 40.0 $\pm0.6$ |
| | PRR | 70.8 $\pm1.1$ | **73.8** $\pm$ NA | 72.2 $\pm0.2$ | 73.1 $\pm0.1$ | 72.6 $\pm1.3$ | 72.7 $\pm1.1$ | 62.3 $\pm0.6$ |
| | ECE | 18.3 $\pm0.8$ | **3.8** $\pm$ NA | 14.8 $\pm0.4$ | 7.2 $\pm0.4$ | 14.9 $\pm0.3$ | 7.2 $\pm0.2$ | 39.1 $\pm1.0$ |
| | NLL | 2.15 $\pm0.05$ | **1.51** $\pm$ NA | 1.77 $\pm0.01$ | 1.83 $\pm0.02$ | 1.78 $\pm0.01$ | 1.84 $\pm0.02$ | 2.61 $\pm0.01$ |

---

[1]The auxiliary training data for CIFAR-10 is CIFAR-100, and CIFAR-10 for CIFAR-100/TinyImageNet.

[2]This is a consistent configuration to the EnD$_{+AUX}$ and EnD$^2_{+AUX}$ models, but constitutes a degraded OOD detection baseline for CIFAR-100 and TinyImageNet datasets due to the choice of auxiliary data.

Firstly, we investigate the ability of a single model to retain the ensemble's classification and prediction-rejection (misclassification detection) performance after either Ensemble Distillation (EnD) or Ensemble Distribution Distillation (EnD$^2$), with results presented in table 3 in terms of error rate and *prediction rejection ratio* (PRR), respectively. A higher PRR indicates that the model is able to better detect and reject incorrect predictions based on measures of uncertainty. This metric is detailed in appendix B. Note that for prediction rejection we used the confidence of the max class, which also is a measure of *total uncertainty*, like entropy, but is more sensitive to the prediction (Malinin & Gales, 2018).

Table 3 shows that both EnD and EnD$^2$ are able to retain both the improved classification and prediction-rejection performance of the ensemble relative to individual models trained with maximum likelihood on all datasets, both with and without auxiliary training data. Note, that on C100 and TIM, EnD$^2$ yields marginally better classification performance than EnD. Furthermore, EnD$^2$ also either consistently outperforms or matches EnD in terms of PRR on all datasets, both with and without auxiliary training data. This suggests that EnD$^2$ is able to yield benefits on top of standard Ensemble Distillation due to retaining information about the diversity of the ensemble. In comparison, a Prior Network, while yielding a classification performance between that on an individual model and EnD$^2_{+AUX}$, performs consistently worse than *all* models in terms of PRR and for all datasets. This is likely because of the inappropriate choice of auxiliary training data and auxiliary loss weight (Malinin & Gales, 2019) for this task.

Secondly, it is known that ensembles yield improvements in the calibration of a model's predictions (Lakshminarayanan et al., 2017; Ovadia et al., 2019). Thus, it is interesting to see whether EnD and EnD$^2$ models retain these improvements in terms of test-set negative log-likelihood (NLL) and expected calibration error (ECE). Note that calibration assesses uncertainty quality on a *per-dataset*, rather than *per-prediction*, level. From table 3 we can see that both Ensemble Distillation and Ensemble Distribution Distillation seem to give similarly minor gains in NLL over a single model. However, EnD seems to have marginally better NLL performance, while EnD$^2$ tends to yield better calibration performance. There are seemingly limited gains in ECE and NLL when using auxiliary data during distillation for EnD$^2$, and sometimes even a degradation in NLL and ECE. This may be due to the Dirichlet output distribution attempting to capture non-Dirichlet-distributed ensemble predictions (especially on auxiliary data) and over-estimating the support, and thereby failing to fully reproduce the calibration of the original ensemble. Furthermore, metrics like ECE and NLL are evaluated on in-domain data, which would explain the lack of improvement from distilling ensemble behaviour on auxiliary data. However, all distillation models are (almost) always calibrated as better than individual models. At the same time, the Prior Network models yield *significantly* worse calibration performance in terms of NLL and ECE than *all* other models. For the CIFAR-100 and TIM datasets this may be due to the target being a flat Dirichlet distribution for the auxiliary data (CIFAR-10), which drives the model to be very under-confident.

Table 4: OOD detection performance (mean % AUC-ROC $\pm 2\sigma$) for C10/C100/TIM models using measures of total (T.Unc) and knowledge (K.Unc) uncertainty.

| Train. Data | OOD Data | Unc. | Individual | Ensemble | EnD | EnD$^2$ | EnD$_{+AUX}$ | EnD$^2_{+AUX}$ | PN$_{+AUX}$ |
|---|---|---|---|---|---|---|---|---|---|
| C10 | LSUN | T.Unc | 91.3 ±1.3 | 94.5 ±N/A | 89.0 ±1.3 | 91.5 ±0.8 | 88.6 ±1.1 | 94.4 ±0.7 | 95.7 ±0.9 |
| | | K.Unc | - | 94.4 ±N/A | - | 92.2 ±0.7 | - | 93.8 ±0.7 | **95.8** ±0.8 |
| | TIM | T.Unc | 88.9 ±1.6 | 91.8 ±N/A | 86.9 ±1.2 | 88.6 ±1.5 | 86.5 ±1.6 | 91.3 ±0.8 | 95.7 ±0.7 |
| | | K.Unc | - | 91.4 ±N/A | - | 88.8 ±1.6 | - | 90.6 ±0.7 | **95.8** ±0.7 |
| C100 | LSUN | T.Unc | 75.6 ±1.1 | 82.4 ±N/A | 73.8 ±0.6 | 80.6 ±1.1 | 80.6 ±0.6 | 83.6 ±0.5 | 74.8 ±1.7 |
| | | K.Unc | - | **88.4** ±N/A | - | 83.8 ±0.9 | - | 86.5 ±0.5 | 73.8 ±2.2 |
| | TIM | T.Unc | 70.5 ±1.5 | 76.6 ±N/A | 68.5 ±1.2 | 74.4 ±1.5 | 74.2 ±0.8 | 77.7 ±0.9 | 73.4 ±2.9 |
| | | K.Unc | - | **81.7** ±N/A | - | 77.2 ±1.5 | - | 80.5 ±1.1 | 72.7 ±3.3 |
| TIM | LSUN | T.Unc | 67.5 ±1.3 | 69.7 ±N/A | 68.7 ±0.2 | 69.6 ±1.4 | 68.8 ±0.2 | 69.2 ±0.7 | 63.7 ±0.1 |
| | | K.Unc | - | 69.3 ±N/A | - | **70.4** ±1.3 | - | 70.3 ±0.4 | 59.5 ±1.9 |
| | C100 | T.Unc | 71.7 ±2.5 | 75.2 ±N/A | 73.1 ±0.7 | 74.8 ±0.3 | 73.1 ±0.3 | 74.1 ±1.3 | **100.0** ±0.0 |
| | | K.Unc | - | 78.8 ±N/A | - | 76.7 ±0.5 | - | 75.3 ±0.7 | **100.0** ±0.0 |

Thirdly, Ensemble Distribution Distillation is investigated on the task of out-of-domain (OOD) input detection (Hendrycks & Gimpel, 2016), where measures of uncertainty are used to classify inputs as either in-domain (ID) or OOD. The ID examples are the test set of C10/100/TIM, and the test OOD examples are chosen to be the test sets of LSUN (Yu et al., 2015), C100 or TIM, such that the test OOD data is *never seen* by the model during training. Table 4 shows that the measures of uncertainty derived from the ensemble outperform those from a single neural network. Curiously, standard ensemble distillation (EnD) clearly fails to capture those gains on C10 and TIM, both with and without auxiliary training data, but does reach a comparable level of performance to the ensemble on C100. Curiously, $\text{EnD}_{+AUX}$ performs *worse* than the individual models on C10. This is in contrast to results from (Englesson & Azizpour, 2019; Li & Hoiem, 2019), but the setups, datasets and evaluation considered there are very different to ours. On the other hand, Ensemble Distribution Distillation is generally able to reproduce the OOD detection performance of the ensemble. When auxiliary training data is used, $\text{EnD}^2$ is able to perform on par with the ensemble, indicating that it has successfully learned how the distribution of the ensemble behaves on unfamiliar data. These results suggest that not only is $\text{EnD}^2$ able to preserve information about the diversity of an ensemble, unlike standard EnD, but that this information is important to achieving good OOD detection performance. Notably, on the CIFAR-10 dataset PNs yield the best performance. However, due to generally inappropriate choice of OOD (auxiliary) training data, PNs show inferior (with one exception) OOD detection performance on CIFAR-100 and TIM.

Curiously, the ensemble sometimes displays better OOD detection performance using measures of *total uncertainty*. This is partly a property of the in-domain dataset - if it contains a small amount of data uncertainty, then OOD detection performance using total uncertainty and knowledge uncertainty should be almost the same (Malinin & Gales, 2018; Malinin, 2019). Unlike the toy dataset considered in the previous section, where significant data uncertainty was added, the image datasets considered here do have a low degree of data uncertainty. It is important to note that on more challenging tasks, which naturally exhibit a higher level of data uncertainty, we would expect that the decomposition would be more beneficial. Additionally, it is also possible that if different architectures, training regimes and ensembling techniques are considered, an ensemble with a better estimate of knowledge uncertainty can be obtained. Despite the behaviour of the ensemble, $\text{EnD}^2$, especially with auxiliary training data, tends to have better OOD detection performance using measures of *knowledge uncertainty*. This may be due to the use of a Dirichlet output distribution, which may not be able to fully capture the details of the ensemble's behaviour. Furthermore, as $\text{EnD}^2$ is (implicitly) trained by minimizing the *forward* KL-divergence, which is *zero-avoiding* (Murphy, 2012), it is likely that the distribution learned by the model is 'wider' than the empirical distribution. This effect is explored in the next subsection and in appendix C.

## 5.1 Appropriateness of Dirichlet distribution

Throughout this work, a Prior Network that parametrizes a Dirichlet was used for distribution-distilling ensembles of models. However, the output distributions of an ensemble for the same input are not necessarily Dirichlet-distributed, especially in regions where the ensemble is diverse. In the previous section we saw that $\text{EnD}^2$ models tend to have higher NLL than EnD models, and while $\text{EnD}^2$ achieves good OOD detection performance, it doesn't fully replicate the ensemble's behaviour. Thus, in this section, we investigate how well a model which parameterizes a Dirichlet distribution is able to capture the exact behaviour of an ensemble of models, both in-domain and out-of-distribution.

Figure 4 shows histograms of *total uncertainty*, *data uncertainty* and *total uncertainty* yielded by an ensemble, $\text{EnD}^2$ and $\text{EnD}^2_{+AUX}$ models trained on the CIFAR-10 dataset. The top row shows the uncertainty histogram for ID data, and the bottom for *test* OOD data (a concatenation of LSUN and TIM). On in-domain data, $\text{EnD}^2$ is seemingly able to emulate the uncertainty metrics of the ensemble well, though does have a longer tail of high-uncertainty examples. This is expected, as on in-domain examples the ensemble will be highly concentrated around the mean. This behaviour can be adequately modelled by a Dirichlet. On the other hand, there is a noticeable mismatch between the ensemble and $\text{EnD}^2$ in the uncertainties they yield on OOD data. Here, $\text{EnD}^2$ consistently yields higher uncertainty predictions than the original ensemble. Notably, adding auxiliary training data makes the model yield even higher estimates of *total uncertainty* and *data uncertainty*. At the same time, by using auxiliary training data, the model's distribution of *knowledge uncertainty* starts to look more like the ensembles, but shifted to the right (higher).

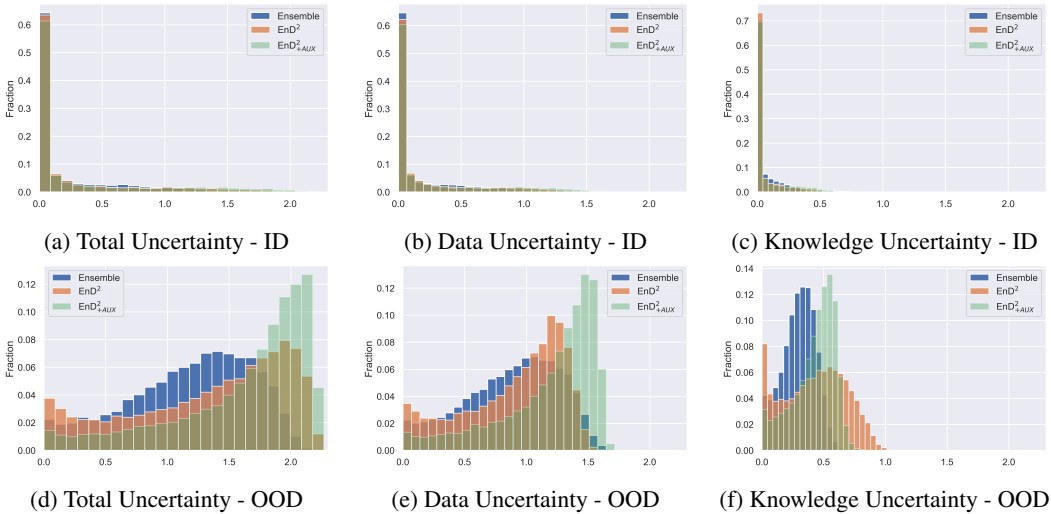

Figure 4: Histograms of uncertainty of the CIFAR-10 ensemble, $EnD^2$ and $EnD^2_{+AUX}$ on *in-domain* (ID) and test *out-of-domain* (OOD) data.

Altogether, this suggests that samples from the ensemble are diverse in a way that's different from a Dirichlet distribution. For instance, the distribution could be multi-modal or crescent-shaped. Thus, as a consequence of this, combined with *forward* KL-divergence between the model and the empirical distribution of the ensemble being *zero-avoiding*, the model over-estimates the support of the empirical distribution, yielding an output which is both more diverse and higher entropy than the original ensemble. This does not seem to adversely impact OOD detection performance as measures of uncertainty for ID and OOD data are further spread apart and the *rank ordering* of ID and OOD data is either maintained or improved, which is supported by results from section 5. However, this does prevent the $EnD^2$ models from fully retaining the calibration quality of the ensemble. It is possible that the ensemble could be better modelled by a different output distribution, such as a *mixture of Dirichlet distributions* or a *Logistic-normal distribution*.

## 6 CONCLUSION

Ensemble Distillation approaches have become popular, as they allow a single model to achieve classification performance comparable to that of an ensemble at a lower computational cost. This work proposes the novel task *Ensemble Distribution Distillation* ($EnD^2$) — distilling an ensemble into a single model, such that it exhibits both the improved classification performance of the ensemble and retains information about its *diversity*. An approach to $EnD^2$ based on using Prior Network models is considered in this work. Experiments described in sections 4 and 5 show that on both artificial data and image classification tasks it is possible to *distribution distill* an ensemble into a single model such that it retains the classification performance of the ensemble. Furthermore, measures of uncertainty provided by $EnD^2$ models match the behaviour of an ensemble of models on artificial data, and $EnD^2$ models are able to differentiate between different *types* of uncertainty. However, this may require obtaining auxiliary training data on which the ensemble is more diverse in order to allow the distribution-distilled model to learn appropriate out-of-domain behaviour. On image classification tasks measures of uncertainty derived from $EnD^2$ models allow them to outperform both single NNs and EnD models on the tasks of misclassification and out-of-distribution input detection. These results are promising, and show that Ensemble Distribution Distillation enables a single model to capture more useful properties of an ensemble than standard Ensemble Distillation. Future work should further investigate properties of temperature annealing, investigate ways to enhance the diversity of an ensemble, consider different sources of ensembles and model architectures, and examine more flexible output distributions, such as mixtures of Dirichlets. Furthermore, while this work considered Ensemble Distribution Distillation only for classification problems, it can and should also be investigated for regression tasks. Finally, it may be interesting to explore combining Ensemble Distribution Distillation with standard Prior Network training.

ACKNOWLEDGEMENTS

This research was partly funded by Cambridge Assessment English via the ALTA Institute, Cambridge University.

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

## APPENDIX A   DATASETS, MODEL ARCHITECTURE AND TRAINING

Table 5: Description of datasets used in the experiments in terms of number of images and classes.

| Dataset | Train | Valid | Test | Classes |
|---|---|---|---|---|
| CIFAR-10 | 50000 | - | 10000 | 10 |
| CIFAR-100 | 50000 | - | 10000 | 100 |
| TinyImagenet | 100000 | - | 10000 | 200 |
| LSUN (evaluation only) | - | - | 10000 | 10 |

All models considered in this work were implemented in Pytorch (Paszke et al., 2017) using a variant of the VGG16 (Simonyan & Zisserman, 2015) architecture for image classification. DNN and EnD models were trained using the negative log-likelihood loss of the labels and the mean ensemble predictions respectively. EnD$^2$ models were trained using the negative log-likelihood of the ensemble's output categorical distributions. All models were trained using the Adam (Kingma & Ba, 2015) optimizer, with a 1-cycle learning rate policy and dropout regularization. For all ensembles, models were trained using different random seed initialization, and using different seeds for shuffling the data. In addition, data augmentation was applied via random left-right flips, random shifts up to $\pm 4$ pixels and random rotations by up to $\pm$ 15 degrees. Tables e 6 details the training configurations for all models. Furthermore, batch normalisation was used for both Ensemble Distillation and Ensemble Distribution Distillation, but not for Prior Networks.

Table 6: Training Configurations. $\eta_0$ is the initial learning rate, $T_0$ is the initial temperature and 'Annealing' refers to whether a temperature annealing schedule was used. The batch size for all models was 128. Dropout rate is quoted in terms of probability of *not* dropping out a unit.

| Training Dataset | Model | General | | | | Distillation | | |
|---|---|---|---|---|---|---|---|---|
| | | $\eta_0$ | Epochs | Cycle len. | Dropout | $T_0$ | Annealing | AUX data |
| CIFAR-10 | DNN | | 45 | 30 | 0.5 | - | - | - |
| | EnD | | 90 | 60 | 0.7 | 2.5 | No | - |
| | EnD$_{+\text{AUX}}$ | $10^{-3}$ | 90 | 60 | 0.7 | 2.5 | No | CIFAR-100 |
| | EnD$^2$ | | 90 | 60 | 0.7 | 10 | Yes | - |
| | EnD$^2_{+\text{AUX}}$ | | 90 | 60 | 0.7 | 10 | Yes | CIFAR-100 |
| | PN | $5\times10^{-4}$ | 45 | 30 | 0.7 | - | No | CIFAR-100 |
| CIFAR-100 | DNN | | 100 | 150 | 0.5 | - | - | - |
| | EnD | | 200 | 150 | 0.9 | 2.5 | No | - |
| | EnD$_{+\text{AUX}}$ | $10^{-3}$ | 200 | 150 | 0.9 | 2.5 | No | CIFAR-10 |
| | EnD$^2$ | | 200 | 150 | 0.9 | 10 | Yes | - |
| | EnD$^2_{+\text{AUX}}$ | | 200 | 150 | 0.9 | 10 | Yes | CIFAR-10 |
| | PN | $5\times10^{-4}$ | 100 | 70 | 0.7 | - | No | CIFAR-10 |
| TinyImageNet | DNN | $10^{-3}$ | 100 | 70 | 0.5 | - | - | - |
| | EnD | $5\times10^{-4}$ | 200 | 150 | 0.8 | 2.5 | No | - |
| | EnD$_{+\text{AUX}}$ | $5\times10^{-4}$ | 200 | 150 | 0.8 | 2.5 | No | CIFAR-10 |
| | EnD$^2$ | $5\times10^{-4}$ | 200 | 150 | 0.8 | 10 | Yes | - |
| | EnD$^2_{+\text{AUX}}$ | $5\times10^{-4}$ | 200 | 150 | 0.8 | 10 | Yes | CIFAR-10 |
| | PN | $5\times10^{-4}$ | 100 | 70 | 0.7 | - | No | CIFAR-10 |

To create the *transfer set* $\mathcal{D}_{\text{ens}}$, ensembles were evaluated on the *unaugmented* CIFAR-10 and CIFAR-100 training examples. During distillation (both EnD and EnD$^2$), models were trained on the augmented examples with the ensemble predictions on the corresponding unaugmented inputs.

## A.1 Numerical issues with maximum likelihood Dirichlet training

As shown in equation 8, ensemble distribution distillation into a prior network via likelihood maximisation is equivalent to minimising the loss function:

$$\mathcal{L}(\phi, \mathcal{D}_{\text{ens}}) = -\frac{1}{N} \sum_{i=1}^{N} \left[ \ln \Gamma(\hat{\alpha}_0^{(i)}) - \sum_{c=1}^{K} \ln \Gamma(\hat{\alpha}_c^{(i)}) + \frac{1}{M} \sum_{m=1}^{M} \sum_{c=1}^{K} (\hat{\alpha}_c^{(i)} - 1) \ln \pi_c^{(im)} \right] \quad (11)$$

If, due to numerical precision of the implementation, one of the ensemble member sample terms $\pi_c^{(im)}$ gets rounded to $0.0$, the log term $\ln \pi_c^{(im)}$ in the above equation cannot be computed. To avoid this, we apply a small amount of *central smoothing* to the ensemble predictions:

$$\pi_{c,smoothed}^{(im)} = (1 - \gamma)\pi_c^{(im)} + \gamma \frac{1}{K} \quad (12)$$

Where the smoothing parameter parameter $\gamma$ has been set to $1 \times 10^{-4}$ for all EnD$^2$ experiments. Note that after applying central smoothing, each ensemble member's predictions still sum up to $1.0$.

## A.2 Temperature Annealing Schedule

A fixed temperature of $2.5$ was used for Ensemble Distillation as recommended in (Hinton et al., 2015), and was found to yield the best classification performance out of $\{1, 2.5, 5, 10\}$. *Temperature annealing* resulted in worse classification performance for Ensemble Distillation, and hence was not used in the experiments. For the temperature annealing schedule, the temperature was kept fixed to initial temperature $T_0$ for the first half-cycle. For the second half-cycle we linearly decayed the initial temperature $T_0$ down to $1.0$. Then, the temperature was kept constant at $1.0$ for the remainder of the epochs. For Ensemble Distribution Distillation, we found that an initial temperature of $10$ performed best out of $\{5, 10, 20\}$. The choice of the particular annealing schedule used has not been tested extensively, and it is possible that other schedules that lead to faster training exist. However, we must point out that it was necessary to use temperature scaling to train models on the CIFAR-10, CIFAR-100 and TinyImageNet datasets.

## Appendix B   Assessing misclassification detection performance

In this work measures of uncertainty are used for two practical applications of uncertainty - misclassification detection and out-of-distribution sample detection. Both can be seen as an outlier detection task based on measures of uncertainty, where misclassifications are one form of outlier and out-of-distribution inputs are another form of outlier. These tasks can be formulated as threshold-based binary classification (Hendrycks & Gimpel, 2016). Here, a detector $\mathcal{I}_T(\boldsymbol{x})$ assigns the label 1 (uncertain prediction) if an uncertainty measure $\mathcal{H}(\boldsymbol{x})$ is above a threshold $T$, and label 0 (confident prediction) otherwise. This uncertainty measure can be any of the measures discussed in sections 2 and 3.

$$\mathcal{I}_T(\boldsymbol{x}) = \begin{cases} 1, & \mathcal{H}(\boldsymbol{x}) > T \\ 0, & \mathcal{H}(\boldsymbol{x}) \leq T \end{cases} \quad (13)$$

Given a set of true positive examples $\mathcal{D}_{\text{p}} = \{\boldsymbol{x}_p^{(i)}\}_{i=1}^{N_p}$ and a set of true negative examples $\mathcal{D}_{\text{n}} = \{\boldsymbol{x}_n^{(j)}\}_{j=1}^{N_n}$ the performance of such a detection scheme can be evaluated at a particular threshold value $T$ using the *true positive rate* $t_p(T)$ and the *false positive rate* $f_p(T)$:

$$t_p(T) = \frac{1}{N_p} \sum_{i=1}^{N_p} \mathcal{I}_T(\boldsymbol{x}_p^{(i)}) \qquad f_p(T) = \frac{1}{N_n} \sum_{j=1}^{N_n} \mathcal{I}_T(\boldsymbol{x}_n^{(j)}) \quad (14)$$

The range of trade-offs between the true positive and the false positive rates can be visualized using a Receiver-Operating-Characteristic (ROC) and the quality of the possible trade-offs can be summarized using the area under the ROC curve (AUROC) (Murphy, 2012). If there are significantly more negatives than positives, however, this measure will over-estimate the performance of the model and yield a high AUROC value (Murphy, 2012). In this situation it is better to calculate the *precision*

and *recall* of this detection scheme at every threshold value and plot them against each other on a Precision-Recall (PR) curve (Murphy, 2012). The recall $R(T)$ is equal to the true positive rate $t_p(T)$, while precision measures the number of true positives among all samples labelled as positive:

$$P(T) = \frac{\sum_{i=1}^{N_p} \mathcal{I}_T(\boldsymbol{x}_p^{(i)})}{\sum_{i=1}^{N_p} \mathcal{I}_T(\boldsymbol{x}_p^{(i)}) + \sum_{j=1}^{N_n} \mathcal{I}_T(\boldsymbol{x}_n^{(j)})} \qquad R(T) = \frac{1}{N_p} \sum_{i=1}^{N_p} \mathcal{I}_T(\boldsymbol{x}_p^{(i)}) \qquad (15)$$

The quality of the trade-offs can again be summarized via the area under the PR curve (AUPR). For both the ROC and the PR curves an ideal detection scheme will achieve an AUC of 100%. A completely random detection scheme will have an AUROC of 50% and the AUPR will be the ratio of the number of positive examples to the total size of the dataset (positive and negative) (Murphy, 2012). Thus, the recall is given by the error rate of the classifier. This makes it difficult to compare different models with different base error rates, as AUPR can increase both due to better misclassification detection and worse error rates.

Below we provide AUPR numbers for all models and datasets in order to illustrate how it can challenging to compare models using this metric. The difference in AUPR between models within a dataset is typically similar to the difference in classification performance. Thus, it is challenging to assesses whether a higher AUPR is due to better misclassification detection or higher error rate. Additionally, this table illustrate that confidence of the prediction is a better measure of uncertainty than entropy for this task. At the same, *knowledge uncertainty* yield much worse misclassification detection performance. These results are consistent with (Malinin & Gales, 2018; Malinin, 2019).

In this work we consider an alternative to using AUPR to assess misclassification detection performance proposed in (Malinin, 2019). Consider the task of misclassification detection - ideally we would like to detect all of the inputs which the model has misclassified based on a measure of

Table 7: Mean Misclassification detection using AUPR using different measures of uncertainty for C10/C100/TIM across three models $\pm 2\sigma$.

| Dataset | Model | Total Uncertainty Confidence | Entropy | Knowledge Uncertainty | % Error |
|---------|-------|------------------------------|---------|-----------------------|---------|
| C10 | DNN | $48.2_{\pm 2.7}$ | $47.0_{\pm 3.4}$ | - | $8.0_{\pm 0.4}$ |
| | ENS | $43.9_{\pm \text{NA}}$ | $41.1_{\pm \text{NA}}$ | $36.8_{\pm \text{NA}}$ | $\mathbf{6.2}_{\pm \text{NA}}$ |
| | EnD | $44.6_{\pm 3.3}$ | $44.1_{\pm 3.9}$ | $37.6_{\pm 2.7}$ | $6.7_{\pm 0.3}$ |
| | EnD$^2$ | $46.8_{\pm 1.2}$ | $46.1_{\pm 1.0}$ | $43.6_{\pm 0.9}$ | $7.3_{\pm 0.2}$ |
| | EnD$_{+AUX}$ | $44.5_{\pm 2.0}$ | $43.8_{\pm 1.7}$ | $37.7_{\pm 1.1}$ | $6.7_{\pm 0.2}$ |
| | EnD$^2_{+AUX}$ | $46.5_{\pm 3.9}$ | $44.3_{\pm 3.5}$ | $39.4_{\pm 3.0}$ | $6.9_{\pm 0.2}$ |
| | PN-RKL | $40.5_{\pm 3.0}$ | $38.5_{\pm 2.7}$ | $35.0_{\pm 2.3}$ | $7.5_{\pm 0.3}$ |
| C100 | DNN | $69.8_{\pm 1.5}$ | $69.4_{\pm 1.4}$ | - | $30.4_{\pm 0.3}$ |
| | ENS | $67.2_{\pm \text{NA}}$ | $64.0_{\pm \text{NA}}$ | $57.6_{\pm \text{NA}}$ | $\mathbf{26.3}_{\pm \text{NA}}$ |
| | EnD | $68.1_{\pm 0.8}$ | $67.6_{\pm 1.1}$ | $62.2_{\pm 1.8}$ | $28.0_{\pm 0.4}$ |
| | EnD$^2$ | $68.3_{\pm 1.6}$ | $66.8_{\pm 1.5}$ | $63.4_{\pm 1.2}$ | $27.9_{\pm 0.3}$ |
| | EnD$_{+AUX}$ | $69.3_{\pm 0.3}$ | $66.9_{\pm 0.2}$ | $58.0_{\pm 0.2}$ | $28.2_{\pm 0.3}$ |
| | EnD$^2_{+AUX}$ | $68.9_{\pm 0.4}$ | $66.7_{\pm 0.2}$ | $62.0_{\pm 1.0}$ | $28.0_{\pm 0.5}$ |
| | PN-RKL | $60.1_{\pm 1.8}$ | $58.0_{\pm 1.8}$ | $52.8_{\pm 2.2}$ | $28.0_{\pm 0.7}$ |
| TIM | DNN | $77.9_{\pm 1.3}$ | $78.3_{\pm 1.2}$ | - | $41.8_{\pm 0.6}$ |
| | ENS | $76.5_{\pm \text{NA}}$ | $74.8_{\pm \text{NA}}$ | $71.8_{\pm \text{NA}}$ | $\mathbf{36.6}_{\pm \text{NA}}$ |
| | EnD | $76.6_{\pm 0.5}$ | $77.3_{\pm 0.8}$ | $70.5_{\pm 1.1}$ | $38.3_{\pm 0.2}$ |
| | EnD$^2$ | $77.0_{\pm 1.0}$ | $76.4_{\pm 0.9}$ | $74.8_{\pm 1.3}$ | $37.6_{\pm 0.2}$ |
| | EnD$_{+AUX}$ | $76.9_{\pm 1.8}$ | $77.1_{\pm 2.1}$ | $69.9_{\pm 1.2}$ | $38.5_{\pm 0.3}$ |
| | EnD$^2_{+AUX}$ | $76.4_{\pm 1.7}$ | $75.8_{\pm 1.6}$ | $74.3_{\pm 2.1}$ | $37.3_{\pm 0.5}$ |
| | PN-RKL | $73.0_{\pm 1.2}$ | $71.1_{\pm 1.1}$ | $60.7_{\pm 1.3}$ | $40.0_{\pm 0.6}$ |

uncertainty. Then, the model can either choose to not provide any prediction for these inputs, or they can be passed over or 'rejected' to an oracle (ie: human) to obtain the correct prediction. The latter process can be visualized using a *rejection curve* depicted in figure 5, where the predictions of the model are replaced with predictions provided by an oracle in some particular order based on estimates of uncertainty. If the estimates of uncertainty are 'useless', then, in expectation, the rejection curve would be a straight line from base error rate to the lower right corner. However, if the estimates of uncertainty are 'perfect' and always bigger for a misclassification than for a correct classification, then they would produce the 'oracle' rejection curve. The 'oracle' curve will go down linearly to 0% classification error at the percentage of rejected examples equal to the number of misclassifications. A rejection curve produced by estimates of uncertainty which are not perfect, but still informative, will sit between the 'random' and 'oracle' curves.

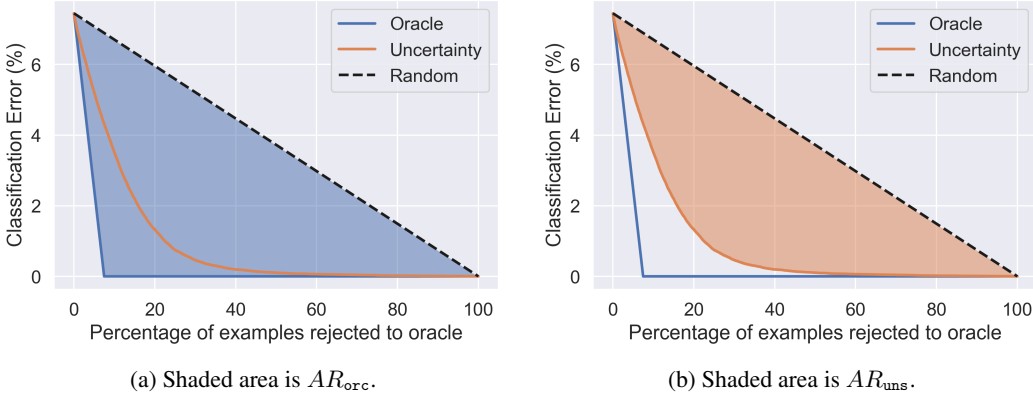

(a) Shaded area is $AR_{\mathrm{orc}}$.          (b) Shaded area is $AR_{\mathrm{uns}}$.

Figure 5: Prediction Rejection Curves

The quality of the rejection curve can be assessed by considering the *ratio* of the area between the 'uncertainty' and 'random' curves $AR_{\mathrm{uns}}$ (orange in figure 5) and the area between the 'oracle' and 'random' curves $AR_{\mathrm{orc}}$ (blue in figure 5). This yields the *prediction rejection area ratio $PRR$*:

$$PRR = \frac{AR_{\mathrm{uns}}}{AR_{\mathrm{orc}}} \qquad (16)$$

A rejection area ratio of 1.0 indicates optimal rejection, a ratio of 0.0 indicates 'random' rejection. A negative rejection ratio indicates that the estimates of uncertainty are 'perverse' - they are higher for accurate predictions than for misclassifications. An important property of this performance metric is that it is independent of classification performance, unlike AUPR, and thus it is possible to compare models with different base error rates. Note, that similar approaches to assessing misclassification detection were considered in (Lakshminarayanan et al., 2017; Malinin et al., 2017)

## APPENDIX C APPROPRIATENESS OF DIRICHLET DISTRIBUTION

Section 5.1 explored the behaviour of measures of uncertainty predicted by models trained on the CIFAR-10 dataset. In this appendix we provide the same for models trained on CIFAR-100 and TinyImageNet. In general, the trends detailed in section 5.1 hold for these datasets as well - EnD$^2$ consistently yield higher uncertainties than the original ensemble, likely as a consequence of the limitations of the Dirichlet distribution.

In addition, we have also provided histograms of *total uncertainty* for EnD and EnD$_{+\mathrm{AUX}}$ models trained on CIFAR-10, CIFAR-100 and TinyImageNet relative to the ensemble for both in-domain and OOD data. Figure 7 shows that while all EnD models match the uncertainty of the ensemble on in-domain data for all datasets, EnD consistently under-estimates the uncertainty for OOD data. The only exception is EnD$_{+\mathrm{AUX}}$ on CIFAR-100, where it matches the ensemble's predictions well. This generally agrees with the results from section 5, where EnD$_{+\mathrm{AUX}}$ is able to almost match the ensemble's OOD detection performance.

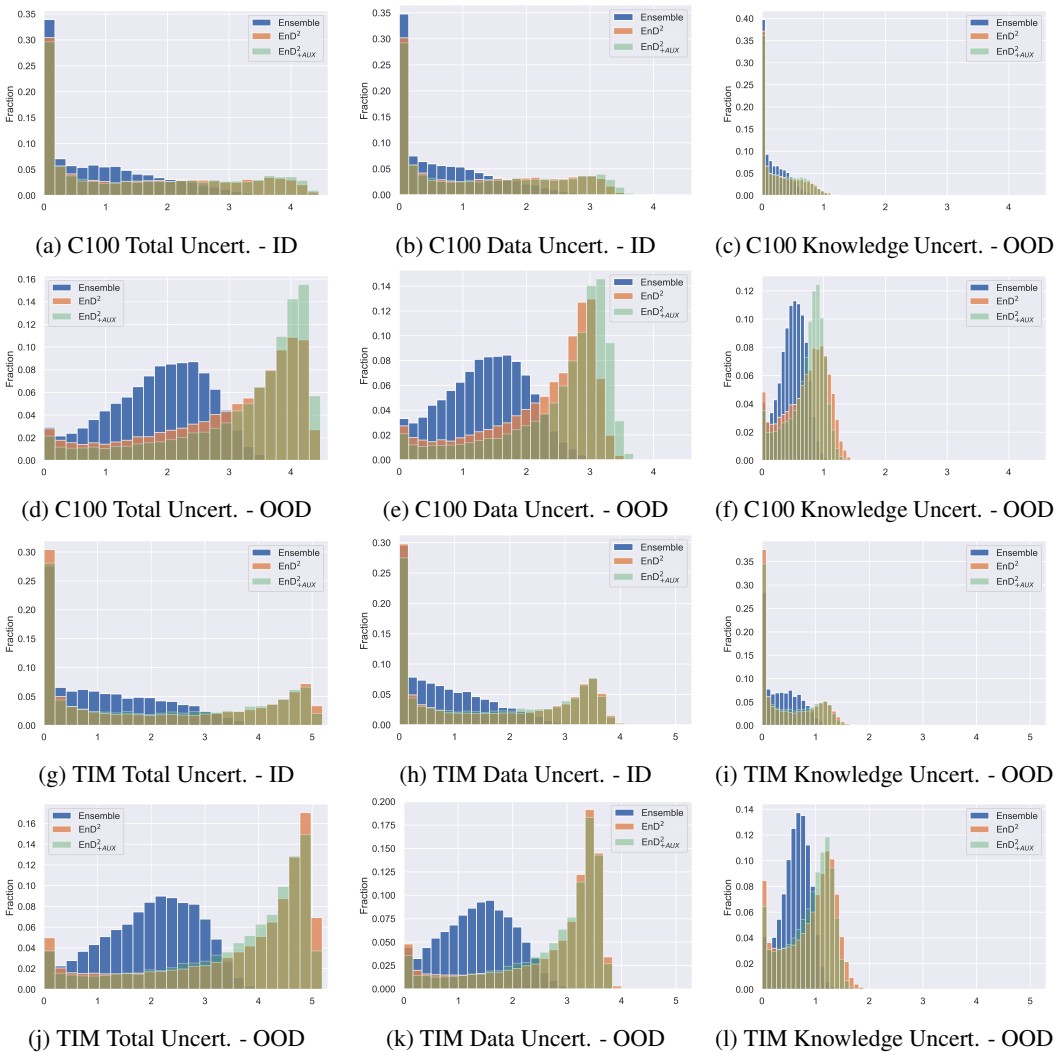

Figure 6: Histograms of measures of uncertainty derived from ensemble, $\text{EnD}^2$ and $\text{EnD}^2_{+\text{AUX}}$ on *in-domain* (ID) and test *out-of-domain* (OOD) data from CIFAR-100 and TinyImageNet.

## APPENDIX D   ABLATION STUDIES

In this paper we made several important design choices. Firstly, we chose to use rich ensembles of 100 models, in order to generate a good estimate of the empirical distribution of the ensemble on the simplex and assess whether the Dirichlet distribution was flexible enough to model it. Secondly, we used a temperature annealing schedule with an initial temperature of 10. In this section we conduct two ablation studies. Firstly, we assess whether Ensemble Distribution Distillation is possible when using an ensemble of 5, 20, 50 and 100 models. Secondly, we investigate a range of initial temperatures (1, 2, 5, 10, and 20) for temperature annealing for the models trained on the CIFAR-10 and CIFAR-100 datasets. Here, only the $\text{EnD}^2_{+AUX}$ model is considered. Results are presented in figures 8-13. All figures depict the mean across 3 models (for each configuration) $\pm$ 1 standard deviation.

Overall, the results show two important trends. Firstly, using using 20 models does better than using 5 models, but there are no conclusive gains for ensemble distribution distillation from using more than 20 models. This suggests that it is sufficient to use ensemble of fewer models for Ensemble Distribution Distillation with models which parameterize the Dirichlet Distribution. This is beneficial in terms computation and memory savings. It is possible, however, that if a more flexible distribution

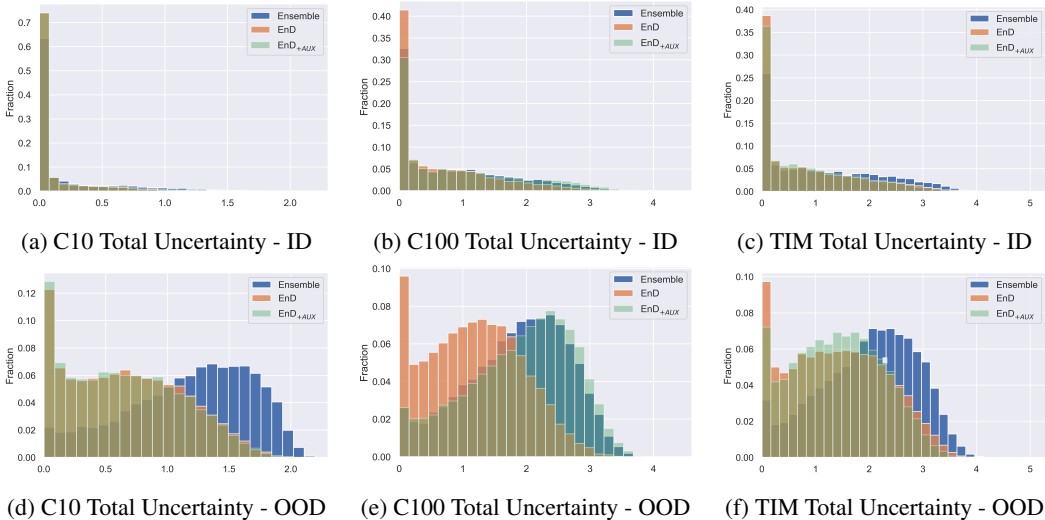

Figure 7: Histograms of measures of total uncertainty derived from ensemble EnD, and EnD$_{+\text{AUX}}$ on *in-domain* (ID) and test *out-of-domain* (OOD) data.

is used, such as a Mixture of Dirichlets, then it might be possible to derive further gains from a larger ensemble. The second trend is that it is necessary to use a temperature of at least 5 in order to successfully distribution-distill the ensemble. Using initial temperatures of 10 and 20 did not result in any significant further increase in performance. These results show that the temperature annealing process is important for Ensemble Distribution Distillation to work well.

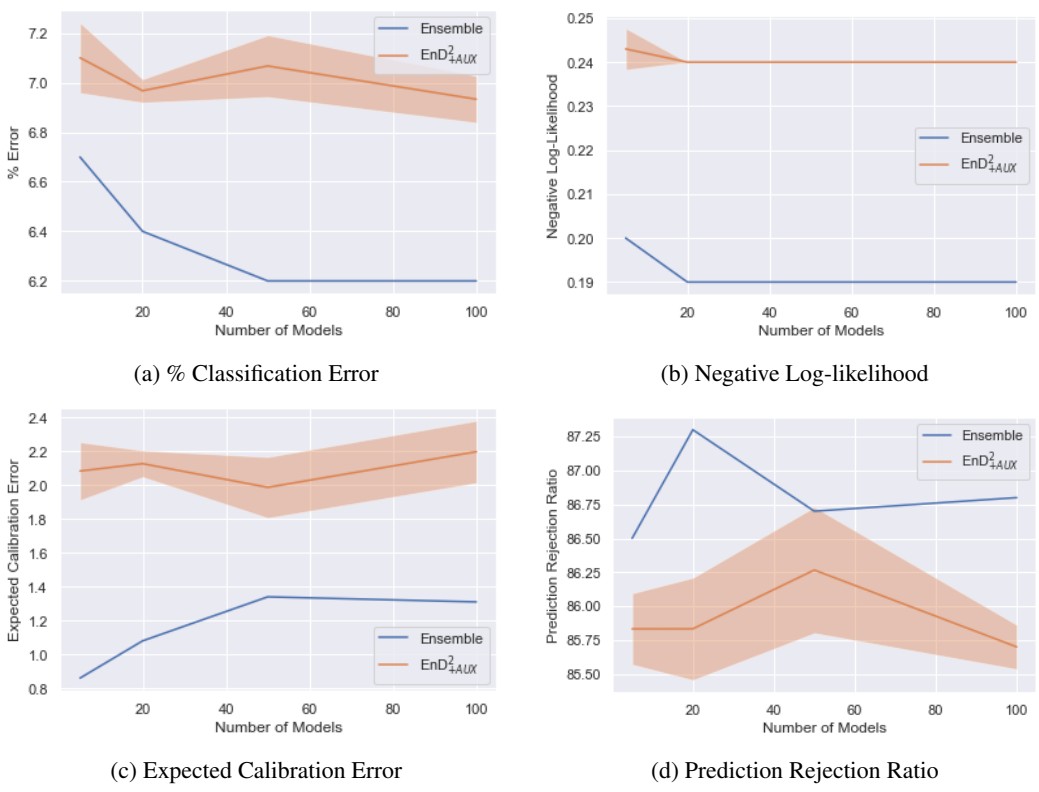

Figure 8: CIFAR-10 Model Ablation

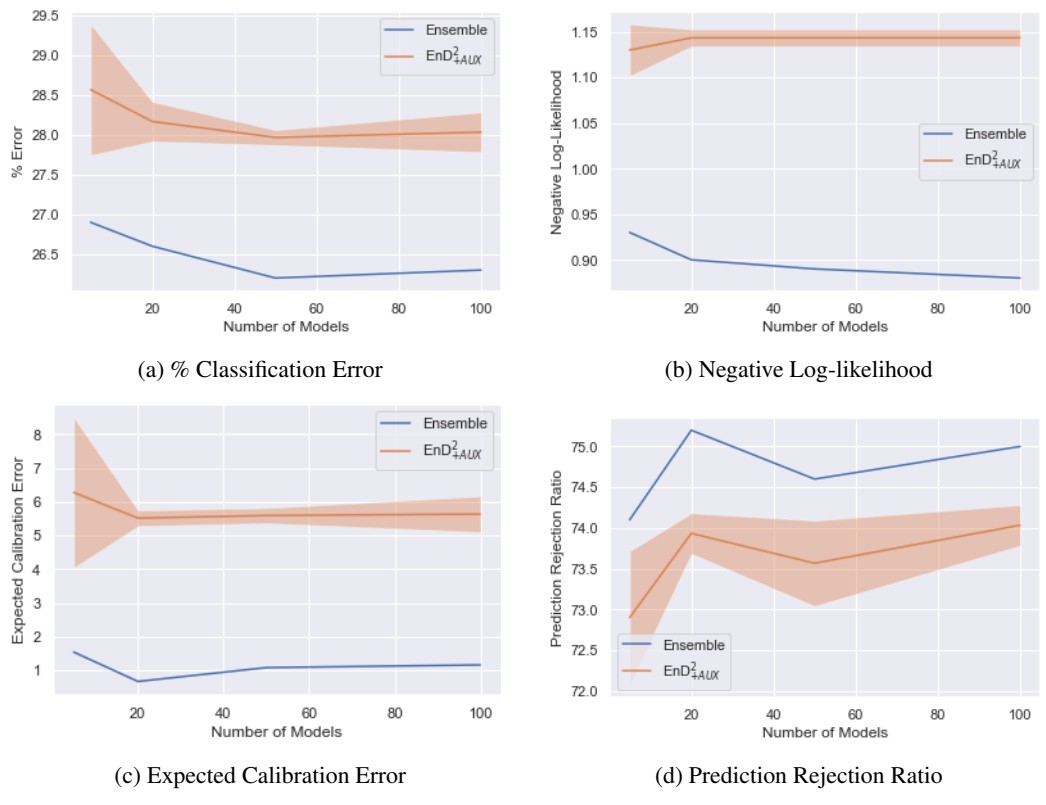

(a) % Classification Error

(b) Negative Log-likelihood

(c) Expected Calibration Error

(d) Prediction Rejection Ratio

Figure 9: CIFAR-100 Model Ablation

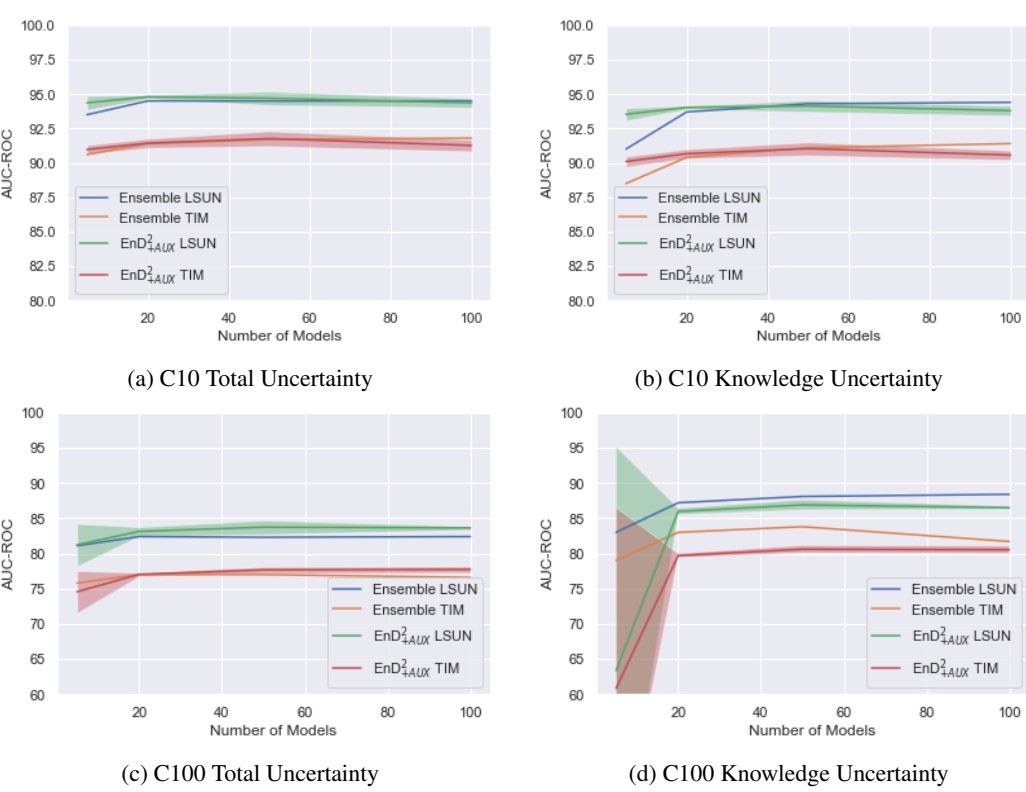

(a) C10 Total Uncertainty

(b) C10 Knowledge Uncertainty

(c) C100 Total Uncertainty

(d) C100 Knowledge Uncertainty

Figure 10: CIFAR-10 and CIFAR-100 Model Ablation - Uncertainties

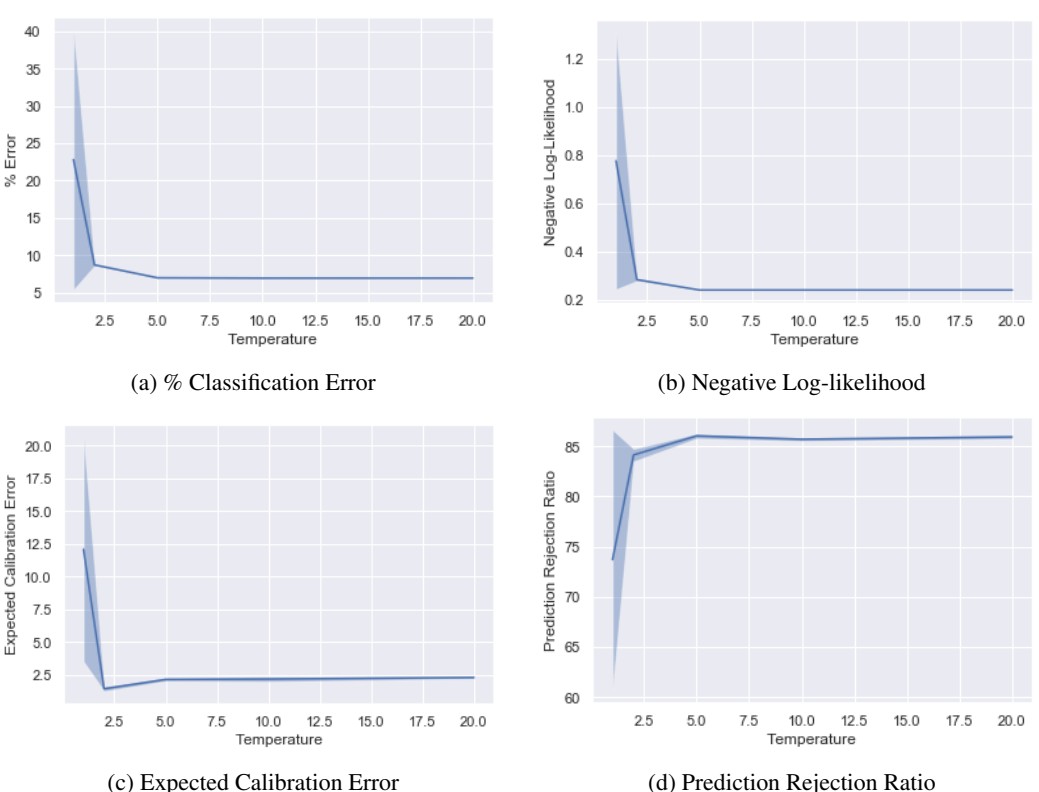

(a) % Classification Error

(b) Negative Log-likelihood

(c) Expected Calibration Error

(d) Prediction Rejection Ratio

Figure 11: CIFAR-10 Temperature Ablation

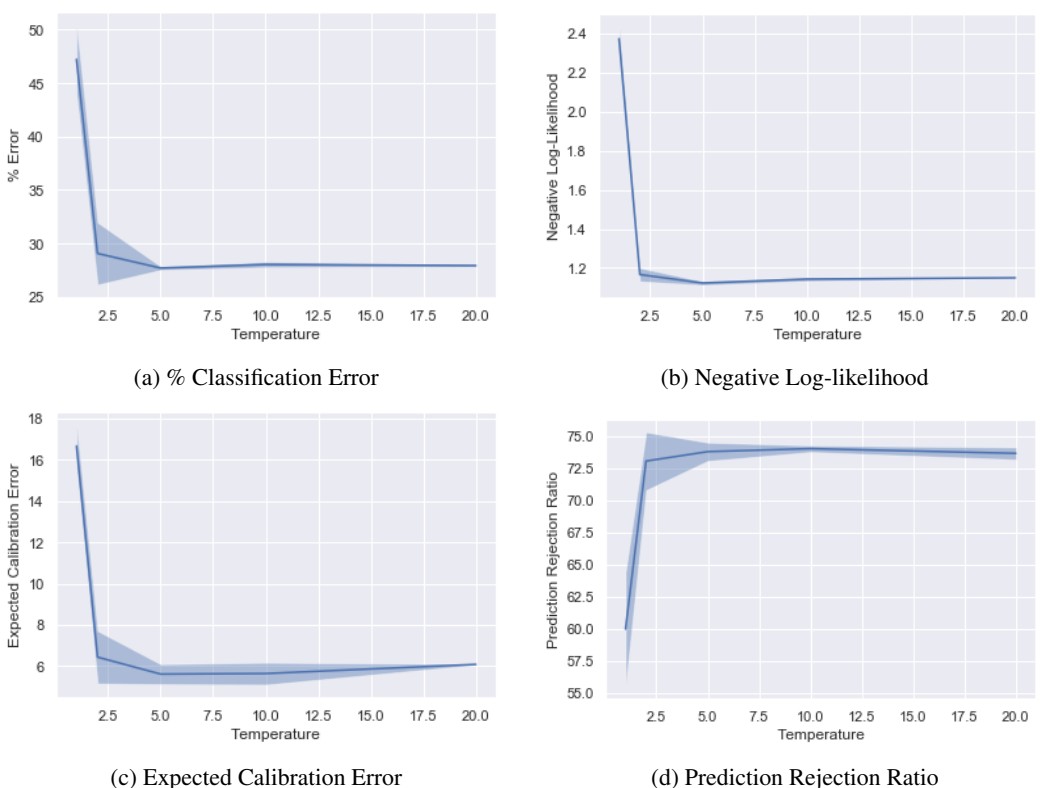

(a) % Classification Error

(b) Negative Log-likelihood

(c) Expected Calibration Error

(d) Prediction Rejection Ratio

Figure 12: CIFAR-100 Temperature Ablation

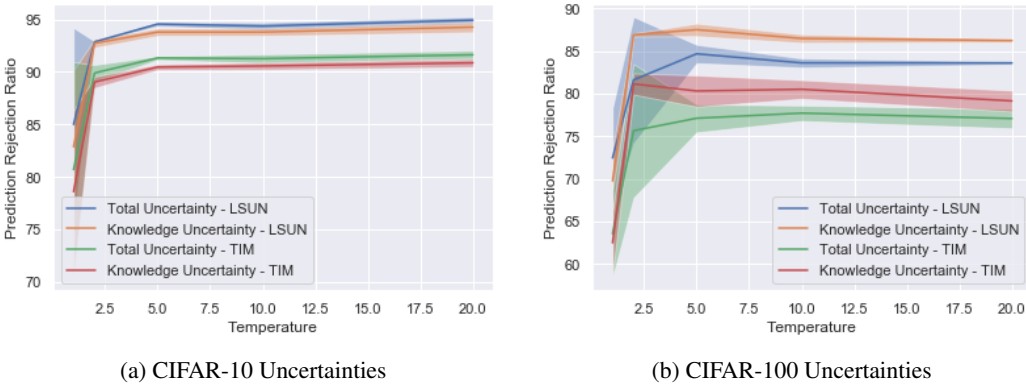

(a) CIFAR-10 Uncertainties

(b) CIFAR-100 Uncertainties

Figure 13: CIFAR-10 and CIFAR-100 Temperature Ablation - Uncertainties

