# OpenReview forum: "Ensemble Distribution Distillation"
_ICLR.cc/2020/Conference — Accept (Poster)_

### Official Review · AnonReviewer1 · 2019-10-17
**Official Blind Review #1**

**Rating:** 8

**Review:**

Summary:

Ensembles of probabilistic models (e.g. for classification) provide measures of both data uncertainty (i.e. how uncertain each model is on average) and model uncertainty (i.e. how much the models disagree in their predictions). When naively distilling an ensemble to a single model, the ability to decompose total uncertainty into data uncertainty and model uncertainty is lost. This paper describes a method for ensemble distillation that retains both types of uncertainty in the case of classification. The idea is to train a prior network (i.e. a conditional Dirichlet distribution) to model the distribution of categorical probabilities within the ensemble. Both total uncertainty and data uncertainty can then be computed analytically from the prior network.

Pros:

The paper considers an interesting problem, and makes a clear contribution towards addressing it. The paper motivates the problem well, and explains the contribution and the method's limitations clearly.

The proposed method is simple, but well motivated, sound, and well explained.

The paper is very well written and easy to read. I particularly appreciated the toy experiment in section 4 and the visualization in figure 3, which showcase clearly what the method does.

The experiments are thorough, and the results are discussed fairly and objectively.

Cons:

The scope of the paper and the method is limited to the problem of probabilistic classification. However, one could have a more general ensemble of conditional or unconditional distributions. The method could in principle be applied to this setting, by having a hypernetwork learn the distribution of the distributional parameters of the models in the ensemble (the method presented in the paper is a special case of this, where the hypernetwork is a prior network and the distributional parameters are categorical probabilities). However, it's not clear that the method would scale to models with arbitrary distributional parameters. I would suggest that the paper make it clear from the beginning that the scope is probabilistic classification, and at the end discuss the extent to which this is a limitation, and how the method could potentially be extended to other kinds of ensembles.

The prior network used is essentially a conditional Dirichlet distribution, which (as the paper clearly acknowledges) may not always be flexible enough. A more flexible prior network could be a mixture of Dirichlet distributions, where the mixing coefficients and the parameters of each mixture component would be functions of the input, similarly to mixture density networks (http://publications.aston.ac.uk/id/eprint/373/) but with Dirichets instead of Gaussians. I believe that equations (9) and (10) would still be tractable in that case, as it's tractable to compute expectations under mixtures if the expectations under mixture components are tractable.

One limitation of the approach is that the prior network may not give accurate predictions for inputs it hasn't been trained on (as the paper discusses and section 4 clearly demonstrates). It's not clear how this problem can be overcome in general, and further research may be needed in that direction.

Some of the results in Table 4 are puzzling (as the paper also acknowledges). In particular, the EnD model should be able to retain the performance of the ensemble when using total uncertainty but it doesn't. Also, using knowledge uncertainty doesn't always seem to be better than using total uncertainty, which to some extent defeats the purpose of the method (at least in this particular example). It would be good to investigate further these results. In any case, I appreciate that the paper acknowledges these results, but avoids unjustified speculation about what may be causing them.

Decision:

Overall, this is good work and I'm happy to recommend acceptance. There are some limitations to the method, but these can be seen as motivation for future work.

Minor suggestions for improvement:

This older work is relevant and could be cited (but there is no obligation to do so).
On compressing ensembles:
- Model compression, https://dl.acm.org/citation.cfm?id=1150464
- Compact approximations to Bayesian predictive distributions, https://dl.acm.org/citation.cfm?id=1102457
- Distilling model knowledge, https://arxiv.org/abs/1510.02437
On information-theoretic measures for decomposing total uncertainty as in eq. (4):
- Decomposition of uncertainty in Bayesian deep learning for efficient and risk-sensitive learning, https://arxiv.org/abs/1710.07283

"[Knowledge uncertainty] arises when the test input comes from a different distribution than the one that generated the training data"
This is only one way knowledge uncertainty can arise, it could also arise when the test input comes from the same distribution that generated training data, but there aren't enough training data available.

Some parentheses are dangling at the bottom of page 2, top of page 3 and middle of page 4.

In figure 1, it'd be good to make clear in the caption that the figure is illustrating the simplex of categorical probabilities.

The last paragraph of section 2 uses the term "posterior" repeatedly to refer to P(y|x, \theta) which is confusing. I would call P(y|x, \theta) the "model prediction" or something like that.

Eq. (5) should be without the negative sign I think.

In section 3, I would use a different symbol (e.g. \phi) to denote the parameters of the prior network, to clearly distinguish them from the parameters of the models in the ensemble.

"Optimization of the KL-divergence between distributions with limited non-zero common support is particularly difficult"
Some more evidence in support of this claim is needed I think; either explain why or provide a reference that does.

In section 4, the text says that EnD has "a minor degradation in performance" but table 1 seems to show otherwise. Also, the results of EnD in table 1 and table 2 are different, which makes me think there may be a typo somewhere.

Making the references have consistent format and correct capitalization (e.g. DNNs, Bayesian) would make the paper look even more polished.


**Experience Assessment:**

I have published one or two papers in this area.

**Review Assessment: Checking Correctness Of Derivations And Theory:**

I carefully checked the derivations and theory.

**Review Assessment: Checking Correctness Of Experiments:**

I carefully checked the experiments.

**Review Assessment: Thoroughness In Paper Reading:**

I read the paper thoroughly.

---

> ### Author Response · Authors · 2019-11-13
> **RESPONSE TO REVIEWER I**
>
> Dear Reviewer I,
>
> While in this paper we deal with EnDD for classification tasks, it is straightforward to adapt to regression tasks by considering a model which parameterises the Normal-inverse-Wishart (or a mixture of) distribution. Such models are described in [PhD Thesis, Malinin 2019] . This is now mentioned in the current version of the paper. Your suggestion to do EnDD with a hyper-network is interesting, as it might allow extracting measures of uncertainty based on distributions over model parameters, for example, which may provide additional insights. This could be an interesting direction for future work. :)
>
> You are completely correct - we could parameterize a Mixture of Dirichlet Distributions and it would still be possible to obtain closed-form expressions for the loss function and all measures of uncertainty (eq. 9 and 10). We were initially planning to do this in a journal paper follow up, as it would be of minor novelty for a follow-up ICML/NeurIPS/ICLR submission).
>
> Regarding the advantage of knowledge uncertainty over total uncertainty - this is a property of the source ensemble and the in-domain dataset, rather than Ensemble Distribution Distillation. Regarding the ensemble - it is possible that if we consider different architectures, training regimes and ensembling techniques (SWA-gaussian, checkpoint ensembles, etc...) then we may achieve better measures of knowledge uncertainty. Regarding the dataset - if the in-domain dataset contains a small amount of data uncertainty, then OOD detection performance using total uncertainty and knowledge uncertainty should be almost the same. The image datasets considered in this work do have a low degree of data uncertainty (unlike the toy dataset, where significant data uncertainty was added). Hence, as seen in the table, total uncertainty and knowledge uncertainty measures often give very similar performance. It is important to note that on more challenging tasks, which naturally exhibit a higher level of data uncertainty, we would expect that the decomposition would be more beneficial. This has now been addressed in the paper.
>
> We will make the references more consistent and do additional tidying of the paper shortly.
>
> [PhD Thesis, Malinin 2019] "Uncertainty Estimation in Deep Learning with application to Spoken Language Assessment". University of Cambridge

---

### Official Review · AnonReviewer2 · 2019-10-22
**Official Blind Review #2**

**Rating:** 6

**Review:**



========= Post Rebuttal =========


I appreciate the authors' effort in addressing the raised issues. I think the revised paper has higher quality in that the additional ablation studies are very useful to understand the effectiveness of the method and the importance of the hyperparameters. It now also has higher clarity in that the presentation of the work is considerably more detailed in the last revision. The outstanding issue is that the final results are not consistent across datasets, baselines, and metrics which is concerning. But, I think, overall, the paper is pushing on an interesting line of research which is relevant and educational for ICLR audience. Th concerns on consistency and conclusiveness of the results can hopefully be addressed in a journal version of the work as suggested by the authors.

========= Summary =========

Distilling an ensemble of deep networks into a single student network is a common approach to reduce the inference-time computational complexity. In those cases, the paper poses the question of whether it is useful for the student to capture the diversity of ensemble members’ predictions on top of the mean distribution (as in the standard distillation).
The main motivation is that this captured diversity will enable the student network to decompose the total (predictive) uncertainty into data (aleatoric/irreducible) and knowledge (epistemic/model) uncertainty components.
In order to distill the diversity, it proposes to use “prior networks [Malinin&Gales 2018]” to output parameters of a  Dirichlet distribution over the simplex of possible predictive categorical distributions.
It uses one toy dataset as well as 3 small image classification datasets to compare the performance of the proposed method (EnD^2) with a) standard ensemble distillation (EnD), b) ensemble of deep networks, as well as c) an individual deep network. The performance is measured in terms of classification accuracy, and the quality of the predictive uncertainty for in-distribution (ID) and out-of-distribution (OOD) samples.
It shows that for OOD sample detection, the proposed EnD^2 outperforms the standard EnD and bridges the gap to the full ensemble. The classification accuracy and ID uncertainty quality is similar to the standard EnD.


========= Strengths and Weaknesses =========

+ the paper studies the interesting problem of decoupling data/knowledge uncertainty in the commonly-used framework of knowledge distillation. It proposes a simple solution to the problem. So, it is highly relevant for the field.
+ Results in Table 3 suggest significant improvements over the standard distillation (EnD).
+ Figure 3 is pedagogical and intuitive for the data vs knowledge uncertainty decomposition and the effectiveness of auxiliary dataset for this toy problem.

(I) General concerns:
- The technical novelty is limited to the combination of knowledge distillation and prior networks.
- Two closely-related works are not cited:
[Li&Hoiem, “Reducing Over-confident Errors outside the Known Distribution” 2019] uses auxiliary dataset for ensemble distillation.
[Englesson&Azizpour, “Efficient Evaluation-Time Uncertainty Estimation by Improved Distillation”, 2019] addresses the problem of efficient uncertainty estimation using knowledge distillation.

(II) Concerns regarding the experiments
- As also found curious by the authors, it is concerning that the EnD_Aux results are so low for OOD detection. Both [Li&Hoiem 2019] and [Englesson&Azizpour 2019] suggest that the standard distillation can perform similarly to (or even outperform) the individual model and the ensemble using an auxiliary dataset [Li&Hoiem 2019] or teacher label for augmented samples [Englesson&Azizpour 2019]. It should be mentioned that the OOD dataset setup is different across these works.
- Why only 100-ensembles are used for the real-world experiments? Does a “successful” training of the prior network require a large number of samples? This would be important since prior ensemble works (e.g. [Lakshminarayanan et al. 2017] ) hardly improve beyond 5-15 networks. Training 100 networks is very costly which would be an important limitation of this work in case it’s necessary. An ablation study on the number of networks in the ensemble is required for investigating this trade-off.
- Hyperparameter optimization: Appendix A provides the final values for the hyperparameters of each method. However, it is not clear how these were optimized? Was grid search used? What was the range for all the hyperparameters? Is there a validation set or cross validation is used and in each case how is the split done? Is it exactly the same hyper-parameter optimization that is done for EnD and EnD^2?

(III) Missing from experiments
- The main goal/motivation of the paper is to be able to decompose the total uncertainty when distilling an ensemble into a single model. In that respect, richer and more experiments are required to evaluate this ability. Currently, the experiments are focused on showing that EnD^2 outperforms EnD. The toy dataset, qualitatively, evaluates the decomposition quality with interesting results but is not enough to make conclusions for real-world datasets. For instance, as an additional experiment, plots/statistics can be given on data vs knowledge uncertainty for ID vs OOD samples.
- Prior network [Malinin&Gales 2018] in its standard form is an important single-network baseline that should be included for both ID and OOD experiments.
- Temperature scaling is mentioned to be a vital part of the model. As such, it requires a thorough ablation study to see the results with or without it as well as when changing the temperature and the annealing factor. It should be studied from two aspects: accuracy and convergence failure (as mentioned by the paper)
- P.7: “Prediction Rejection Ratio” (PRR) is a measure proposed in this work but it’s only defined in the appendix. That is the only metric that measures the quality of uncertainty for ID samples (Table 3). As such, I believe it’s important to define PRR in the main paper and also further include more standard metrics such as NLL, AUROC, AUPRcurve in table 3 so that some context is given to the newly-proposed PRR.
- Along the same line, it seems NLL is consistently and “more significantly” worse for EnD^2 compared to EnD.

(IV) Missing training and implementation details
- More details should be provided for the training of the student prior network including it’s loss function given a dataset and Dirichlet distribution. This can be obtained from [Malinin&Gales 2018] but it’s important for this work to be self-contained.
- the loss function in equation 8 has log(p(\pi|x;\theta)); is there any numerical issue regarding the Dirichlet distribution and/or the log? If so, how significant and what measures are accordingly taken? Could it be that the temperature scaling is more a way of alleviating these issues as opposed to the shared support of distributions for KL divergence?
- the details of the annealing algorithm is entirely missing.


========= Final Decision =========

The paper addresses a highly relevant problem in a simple (and potentially effective) way. This is great. However, there are several concerns as listed above which altogether makes me lean towards an initial “weak reject” rating. (II) and (III) are more central to this initial rating. I will carefully read the authors rebuttal as well as other reviewers’ comments before I finalize my rating.


========= Minor points =========

general:
- P.2: “[...] limitation of ensembles is that the computational cost of training and inference [...] One solution is to distill [...]“ -> this only remedies the computational complexity of inference and in fact increases the training time.
- P.3: [Malinin&Gales 2018] should be cited for equation 4 and the discussion around it.
- P.5: “Ensemble Distillation (EnD) is able to recover the classification performance [...] with only very minor degradation in performance”. Table 1 does not show any degradation for EnD. It shows some degradation for EnD^2 when 100-ensemble is used.
- P.7: “Note, that on C100 and TIM, EnD2 yields better classification performance than EnD”: almost all of the classification improvements are well within one-std. In the specific case of C100, it’s a stretch to call it a “better classification performance”.
- P.7: how many runs are used to obtain the mean and std reported in table 3 and 4.
- P.7: is the PRR in table 3 calculated using the total or knowledge uncertainty for ensemble and EnD2?

Typo:
- P.2: “. Consider an ensemble of models {P[…]” --> a closing parenthesis is missing.
- P.3: “Each of the models P([...]” --> a closing parenthesis is missing.
- P.3: “ the entropy of the expected and“ -> “the entropy of the expectation”
- P.4: “=\hat{p}(x,\pi)” → “\sim\hat{p}(x,\pi)”
- P3-4.: hat and star seem to have been arbitrarily used for input x, parameters theta, pi and p with most of them undefined.
- P.5: script MI is used for mutual information while in P.4 script I is used.
- P.6: “may require the use additional training” -> use of additional
- P.6: Results of EnD in table 2 does not match table 1.



**Experience Assessment:**

I have published one or two papers in this area.

**Review Assessment: Checking Correctness Of Derivations And Theory:**

I carefully checked the derivations and theory.

**Review Assessment: Checking Correctness Of Experiments:**

I carefully checked the experiments.

**Review Assessment: Thoroughness In Paper Reading:**

I read the paper thoroughly.

---

> ### Author Response · Authors · 2019-11-13
> **RESPONSE TO REVIEWER II - PART I**
>
> Dear Reviewer II,
>
> We address each of your comments point-by-point below.
>
> (I) General concerns:
>
> I.a See response to reviewer III
>
> I.b Works now cited. See II.a
>
> (II) Concerns regarding the experiments
> II.a Thank you for pointing out these papers, as they are relevant to our work and we have now cited them. While work by [Li&Hoiem 2019] considers an augmented dataset, their datasets, training regime and model architecture are completely different to the one in our work, so it is hard to make a direct comparison. Work by [Englesson&Azizpour 2019] is closer in terms of datasets, overall setup and evaluation. However, an important difference is that they use the ensemble's predictions on augmented data obtained via rotations and other standard perturbations, while we use the ensemble's predictions on the original data for augmented data (flips,shifts,rotations). What they do is sensible and expected to give gains on top of what we do. The reason we didn't do this is because it would be prohibitive in terms of memory to save the predictions of an ensemble of 100 models for all possible forms of perturbation. We would also like to point out that the assessment of the measures of uncertainty in these two works is more narrow in scope in comparison to ours, and mostly focused on calibration and test-set NLL (which we also provide). Regarding the worse performance of EnD_AUX relative to the ensemble on OOD detection - this may be an effect of the training regime we considered.
>
> II.b We agree that an ablation study would be informative and are currently running it. Results will be updated as soon as it is complete. Should be done in about a day or two.
>
> II.c Hyper-parameters were 'manually' tuned for accuracy on a small held-out set of 1000 samples. Then models were trained on the full training set using the same hyper-parameter settings for several random seeds. In general we explored 'around' the same setup as in (Malinin and Gales 2018, 2019). Unfortunately, when we did this study we only had access to a limited set of GPUs, so a thorough parameter grid-search was infeasible. We agree that in general, a more thorough investigation of this method on different architectures (ResNet, DenseNet, etc...) using different training regimes and on both a bigger image dataset (ImageNet) and a different task (machine translation/NLP/speech recognition) would be informative. However, we leave that to future work. Specifically, we plan to address this in a journal paper follow-up.
>
> (III) Missing from experiments
>
> III.a Your concern is valid and we agree that it is desirable to assess EnDD on more complex tasks than CIFA10/CIFAR100/TinyImageNet. However, the datasets we use are consistent with those used by other works in this area (Hendrycks and Gimpel 2016, Kimin Lee et all 2018, Malinin and Gales 2018/2019). In future work we plan to assess EnDD on more challenging tasks, such as NMT/ASR, where both the scale and complexity of data is much larger. At the same time, however, we have added histograms of total/data/knowledge uncertainty for ID/OOD inputs to models trained on each dataset (C10/C100/TIM) in the main paper and in appendix in order to analyse the decomposition of uncertainty achievable through EnDD relative to the original ensemble in a little more detail, as well as further analyse the appropriateness of the Dirichlet.
>
> (Lee et all, 2018a) "Confidence-Calibrated Adversarial Training: Towards Robust Models Generalizing Beyond the Attack Used During Training"
> (Lee et all, 2018b) "A Simple Unified Framework for Detecting Out-of-Distribution Samples and Adversarial Attacks"
>
> III.b We include result Prior Networks trained in a matching configuration to EnDD for the CIFAR10, CIFAR100 and TinyImageNet datasets. The results show that Prior Networks can achieve comparable (but smaller) improvements in classification performance as EnDD, due to the regularizing effect of using more data. However, both on CIFAR-10 and CIFAR-100 misclassification detection, test-set negative log-likelihood and calibration ECE performance are drastically worse for a Prior Network on in-domain data than for all other models. On CIFAR-10 Prior Networks achieve best OOD detection results, but on CIFAR-100 and TinyImagenet do not (with one exception). It is necessary to point out that while the setup for Prior Networks on CIFAR-10 is both identical to "Reverse KL-Divergence Training" (Malinin and Gales 2019) and to the EnDD configuration considered here, the setup for the Prior Network on CIFAR-100 is different. Specifically, we used a matched configuration to EnDD, but this represents a degraded baseline for OOD detection relative to (Malinin and Gales 2019), where the TinyImageNet dataset was used as 'OOD training data'.
>
> III.c Ablation study is currently underway. We will update the results as soon as it is completed.

---

> ### Author Response · Authors · 2019-11-13
> **RESPONSE TO REVIEWER II - PART II**
>
> III.d While it is important to describe PRR, we believe that it will distract from the main story if added to the main part of the paper. Thus, we will keep it in the appendix. It must be noted that this measure was also proposed in a number of different publications and previous papers (not cited in the appendix). We didn't have references previously because overleaf died before we could upload a version of the paper with them. However, we are happy to move the NLL/Calibration experiments to the main paper and discuss them there. Additionally, we can provide a PR curve and AUPR numbers in the appendix for further information. However, this has previously been investigated in [PhD Thesis, Malinin 2019] for a range of datasets.
>
> [PhD Thesis, Malinin 2019] "Uncertainty Estimation in Deep Learning with application to Spoken Language Assessment". University of Cambridge
>
> III.e This is likely a consequence of over-estimating the support due to using forward KL-divergence in combination with a Dirichlet distribution, which may not be flexible enough to capture the distribution of the ensembles. EnD doesn't suffer from this, as it only tries to match the mean of the ensemble. We have moved the NLL results to the main paper and addressed this in the discussion. We have moved to discussion about the appropriateness of the Dirichlet Distribution from the appendix into the main paper as it ties in with this concern.
>
>
> (IV) Missing training and implementation details
>
> IV.a Details added to equation 9.
>
> IV.b The only possible issues is if one of the ensemble's predictions is so sharp that it is a one-hot vector. Then $\ln(\pi_c^{(mi)})$ would be a NaN for the pis which are 0. To avoid this we added a very small smoothing term (now described in appendix) to deal with this even without temperature annealing. While this allows EnDD to work on Toy data, it is still necessary to use temperature for the CIFAR-10/CIFAR-100/TinyImageNet datasets. However, the loss should otherwise be stable. One of the reasons we believe that temperature affects the common support, rather than stability of the loss, is because the toy problem is 3-dimensional (in terms of classes). As the dimensionality of the problem increases, common support is harder to achieve due to how high-dimensional spaces operate.
>
> IV.c Details of annealing schedule have been added to the appendix.
>
> (V) Minor Points
>
> V.a,b,c - All addressed in the paper.
>
> V.d 100 DNN models were trained. 3 EnD and EnDD models were run. 10 for C10 Prior Network, 3 for C100 and TIM Prior Network. We initially invested a lot of compute to construct a rich ensemble, and then a lot of time was spent getting EnDD working for image data. However, all models tend to yield roughly consistent performance across different random seeds, so we considered 3 random seeds sufficient. Using more would, of course, be better.
>
> V.e We used confidence of the max class, which is also a measures of total uncertainty. It was shown in (Malinin and Gales, 2018) and in (Phd Thesis, Malinin, 2019) that confidence of the max class consistently yields marginally better performance on misclassification detection than entropy, as it is only sensitive to the probability of the prediction being considered, unlike entropy, which is sensitive to the entire distribution over classes.
>
> TYPOS: All addressed.

---

### Official Review · AnonReviewer3 · 2019-10-24
**Official Blind Review #3**

**Rating:** 6

**Review:**

This paper notes that ensemble distillation (EnD) loses the distributional information from the original ensemble, which prevents users of the distilled model from being able to obtain estimates of its knowledge uncertainty. It proposes the challenge of distilling the ensemble in a way that preserves information about the ensemble distribution, so that knowledge uncertainty can still be extracted from the distilled model. It names this task "Ensemble Distribution Distillation (EnD^2)". It then proposes using Prior Networks (introduced in Malinin & Gales 2018) as a solution, and proceeds to evaluate it with a series of experiments -- first obtaining some intuition from spiral dataset, then more rigorously on benchmark image datasets (CIFAR10/100/TinyImageNet).

First, it trains ensembles of 10 and 100 NNs on a spiral dataset, distills them using the regular approach (EnD) and Prior Networks (EnD^2), compares their performance, and notes that the Prior Networks approach has comparable performance. Next, it visualizes over the input space of the spiral dataset the total uncertainty, data uncertainty, and knowledge uncertainty estimates, which are extracted directly from the original ensemble and from the EnD^2 distilled model (Figure 3). It notes that while the original ensemble is able to correctly estimate the knowledge uncertainty in regions that are far away from the training distribution, the EnD^2 model fails at this task (Figure 3f). It then proposes to augment the training set with out-of-distribution data, and demonstrates that this improves the estimation of knowledge uncertainty (Figure 3i). It also proposes a new metric for evaluating the Prediction Rejection Ratio (PRR), uses it to compare how the EnD^2 model compares to the original ensemble and the EnD model. Finally, it demonstrates using a series of benchmark image classification tasks that the EnD^2 model is able to identify out-of-distribution samples with comparable performance to the original ensemble.

Decision: Leaning-to-Accept. Distillation is a well-established technique, and adapting it so that the same NN can perform both predictions and knowledge uncertainty estimates is impactful. This work proposes using Prior Networks as a way to distill ensembles of NNs in a way that preserves the knowledge uncertainty estimates, and evaluated this claim with a sequence of experiments. This work also proposes a new evaluation metric (Prediction Rejection Ratio), and can be used to evaluate future models that are able to simultaneously perform prediction and knowledge uncertainty estimation. However, the way that the paper is organized around the proposal of "Ensemble Distribution Distillation" as a novel machine learning task does not seem very well motivated, as the distribution was solely used to provide uncertainty estimates.

Strengths:
- The visualizations in Figure 3 helped to provide intuition to the reader.
- Experiments have a clear logical flow. Spiral experiments provide intuition, motivate out-of-distribution data augmentation, then image data experiments provide evidence for the applicability of the method.
- Motivates and explains the newly proposed evaluation metric (prediction rejection ratio) in the appendix.
- The out-of-distribution detection experiments are quite comprehensive.
- The training procedures are clearly detailed in the appendix.
- Investigates the appropriateness of the Dirichlet distribution in the appendix.

Weaknesses:
- The proposal of the novel machine learning task of "Ensemble Distribution Distillation" does not seem very well motivated. In this paper, the distribution distillation was solely used to obtain a knowledge uncertainty estimation. Besides that, what else would the distribution be used for? It was also initially unclear to me what this paper contributes on top of "Predictive Uncertainty Estimation via Prior Networks (Malinin & Gales, 2018)". A suggestion is to rewrite the summary of contributions to emphasize that the use of Prior Networks to produce a single model that can both perform predictions and provide uncertainty estimates as an extension of ensemble distillation is novel, and that a more comprehensive set of experiments on more difficult image datasets were done in this paper.

Minor comments:
- page 2, expression right before equation 2, and in the first sentence on page 3 is missing closing parentheses.
- page 3, figure 1. It wasn't initially obvious to me that the triangle represents the simplex of the softmax output, and each black dot represents the output of one model of the ensemble.
- page 4, equation 9. Add some space to the right of the equality sign.
- Use backticks`   instead of single quotation mark ' to open quotation marks in LaTeX.


Rebuttal response:
I acknowledge the authors' point about the importance of EnDD in addition to knowledge uncertainty estimation, and maintain my rating.

**Experience Assessment:**

I do not know much about this area.

**Review Assessment: Checking Correctness Of Derivations And Theory:**

I did not assess the derivations or theory.

**Review Assessment: Checking Correctness Of Experiments:**

I carefully checked the experiments.

**Review Assessment: Thoroughness In Paper Reading:**

I read the paper thoroughly.

---

> ### Author Response · Authors · 2019-11-13
> **RESPONSE TO REVIEWER III**
>
> Dear Reviewer III,
>
> The benefit of Ensemble Distribution Distillation is the ability to explicitly emulate a source ensemble and decompose measures of uncertainty. This is common practice for tasks such as Bayesian Active Learning (Kirsch 2019), where knowledge uncertainty is commonly used. Thus, we believe that the ability to model an ensemble and decompose uncertainties via EnDD is by itself significant. However, we speculate that Ensemble Distribution Distillation may have additional benefits by consuming extra degrees of freedom from the model and adding a regularizing effect. This may make the model more robust to, for example, adversarial attacks. This is supported by the findings in (Malinin & Gales, 2019), where it is shown that it is more difficult to construct adversarial attacks against Prior Network models due to their rich measures of uncertainty. However, investigating this is left to future work.
>
> (Kirsch, 2019) "BatchBALD: Efficient and Diverse Batch Acquisition for Deep Bayesian Active Learning" (NeurIPS 2019)
> (Malinin and Gales, 2019) "Reverse KL-divergence training of Prior Networks: Improved Uncertainty and Adversarial Robustness" (NeurIPS 2019)

---

### Author Response · Authors · 2019-11-13
**GENERAL RESPONSE TO REVIEWERS**

Dear Reviewers!

We thank you for putting in the time and effort to produce such detailed reviews! We have taken on board your comments, fixed most of the minor errors which you have pointed out, added suggested references and put them in context, clarified certain points and have implemented some of the structural changes which you recommend (though we have now gone over the 8 page suggested limit). A new version of the paper reflecting these changes has been uploaded. The additional experiments suggested by you are under way now. We have added partial results, though we may not be able to complete all the extra experiments within the time-frame of this rebuttal period (but definitely within the next two weeks). We provide a more detailed response to each of you as a comment to each review.

We will inform of any further update to the paper in this comment.

UPDATE:  1. We have added histograms of uncertainty for standard ensemble distillation in the appendix.
                  2. As per reviewer II suggestion, we have also provided AUPR numbers for all models in the appendix. However, they mainly illustrate the AUPR is difficult to use to assess misclassification detection performance.
                  3. We have found a small bug in table 4 - results for EnD and EnD_AUX were carried over from an older table and incorrect. We have updated these numbers to the appropriate model. These results now tie up far better with the new histograms in the appendix and with the paper pointed out to us by reviewer II.

UPDATE II: We have now added the studies of the effects and temperature and number of models in an additional appendix which were requested by reviewer II. In short, the results show that temperature annealing is important, and using an initial temperature of at least 5 is necessary for good ensemble distribution distillation, while a temperature of 10 increases the consistency of the result. Secondly, the results show that using ensemble of 5 models does not allow the ensemble distribution distillation to estimate knowledge uncertainty well, and that using ensembles of 20 models does better. However, not further significant improvements from using more than 20 models are not observed. Overall, this is good, as it suggests that it is possible to do EnDD without having to expensively train a vast ensemble of models.

---

### Decision · Program_Chairs · 2019-12-19

**Decision:**

Accept (Poster)

**Comment:**

The paper investigates how to distill an ensemble effectively (using a prior network) in order to reap the benefits of uncertainty estimation provided by ensembling (in addition to the accuracy gains provided by ensembling).

Overall, the paper is nicely written, and makes a valuable contribution. The authors also addressed most of the initial concerns raised by the reviewers. I recommend the paper for acceptance, and encourage the authors to take into account the reviewer feedback when preparing the final version.